# Pigment epithelium-derived factor promotes peritoneal dissemination of ovarian cancer through induction of immunosuppressive macrophages

Sayaka Ueno [1,2], Tamotsu Sudo[2], Hideyuki Saya [1,3✉] & Eiji Sugihara [1,3✉]

Peritoneal dissemination of ovarian cancer (OC) correlates with poor prognosis, but the mechanisms underlying the escape of OC cells from the intraperitoneal immune system have remained unknown. We here identify pigment epithelium–derived factor (PEDF) as a promoting factor of OC dissemination, which functions through induction of CD206[+] Interleukin-10 (IL-10)–producing macrophages. High PEDF gene expression in tumors is associated with poor prognosis in OC patients. Concentrations of PEDF in ascites and serum are significantly higher in OC patients than those with more benign tumors and correlated with early recurrence of OC patients, suggesting that PEDF might serve as a prognostic biomarker. Bromodomain and extraterminal (BET) inhibitors reduce PEDF expression and limit both OC cell survival and CD206[+] macrophage induction in the peritoneal cavity. Our results thus implicate PEDF as a driver of OC dissemination and identify a BET protein–PEDF–IL-10 axis as a promising therapeutic target for OC.

[1] Division of Gene Regulation, Institute for Advanced Medical Research, Keio University School of Medicine, Shinjuku-ku, Tokyo, Japan. [2] Section of Translational Research, Hyogo Cancer Center, Hyogo, Japan. [3] Division of Gene Regulation, Cancer Center, Research Promotion Headquarters, Fujita Health University School of Medicine, Aichi, Japan. ✉email: hsaya@a5.keio.jp; eiji.sugihara@fujita-hu.ac.jp

Ovarian cancer (OC) is the most lethal disease among all gynecologic malignances, with a 5-year survival rate of 45%[1]. Due to a lack of clinical symptoms during development, ~70% of women with OC are diagnosed at an advanced stage of the disease, at which time the tumor has likely disseminated widely[2]. The peritoneum, including the omentum and pelvic and abdominal viscera, is the most common site for such dissemination[2]. OC is treated with a combination of cytoreductive surgery and platinum-based chemotherapy. Although most OC patients respond to standard therapeutic modalities, most of those who present with advanced disease undergo relapse, with a median time to recurrence of 16 months[3]. The most important prognostic indicator in women with advanced OC is the residual tumor volume after surgical debulking[4,5]. However, the absence of visible residual disease does not mean that no residual disease is present after surgery, as indicated by the fact that ~70% of first relapses are localized to the peritoneal cavity[6,7]. New strategies to treat peritoneal dissemination of OC are thus urgently needed.

The peritoneal cavity is an immunologically complex environment that includes immune aggregates in the peritoneal wall, mesentery, and omentum as well as free immune cells in the peritoneal fluid[8]. This fluid, which facilitates frictionless movement of abdominal organs, also distributes various immune cell subsets throughout the peritoneal cavity. Immune cells in the peritoneal fluid are comprised of mostly monocytes, macrophages, and B1 lymphocytes[8–10], all of which contribute to the elimination of pathogens. However, how OC cells escape from such intraperitoneal immune surveillance is unclear. Characterization of the underlying mechanisms within the immune context might inform the development of new treatment strategies for patients with OC.

Pigment epithelium-derived factor (PEDF) is a 50-kDa secreted glycoprotein that belongs to the family of serine protease inhibitors and was initially isolated from medium conditioned by cultured retinal pigment epithelial cells[11,12]. PEDF exerts a range of biological effects associated with many physiological and pathophysiological processes[13,14]. It has been found to have antiangiogenic, antitumorigenic, and antimetastatic roles in several types of cancer, including pancreatic carcinoma, melanoma, prostate cancer, and glioma[15–18]. Its expression was shown to be higher in hepatocellular carcinoma tissue than in adjacent normal tissue, and its concentration was higher in serum samples from patients with such tumors than in those from control subjects[19,20]. Circulating PEDF levels were also found to be significantly higher in patients with gastric cancer than in those with gastric precancerous lesions or in healthy individuals[21]. In addition, PEDF was shown to promote the stemness and self-renewal properties of glioma stem cells[22]. These various observations indicate that the functions of PEDF are diverse, complicated, and site-specific, with its role in OC remaining unknown.

In the present study, we show that PEDF promotes OC dissemination through induction of CD206[+] Interleukin-10 (IL-10)-producing macrophages in the peritoneal cavity in mouse models. This protumor role of PEDF was indicated in humans where elevated serum and ascites PEDF levels in OC patients correlated with early OC recurrence. BET inhibitors suppress PEDF expression and limit both OC cell survival and induction of immunosuppressive macrophages in the peritoneal cavity. Together, this study highlights the important roles of PEDF in OC dissemination, the potential application of serum PEDF as a biomarker to predict worse prognosis, and novel promising therapeutic target for OC, that is, a BET protein-PEDF–IL-10 axis.

## Results

**Establishment of a highly disseminating mouse OC cell line.** To characterize the mechanisms by which OC cells disseminate into the peritoneal cavity, we established a new cell line through in vivo selection of ID8 mouse OC cells that express GFP (ID8G cells). Tumors and ascites fluid developed in syngeneic C57BL/6 J mice 3–4 months after i.p. injection of ID8G cells. Cells isolated from omental tumors by FACS were termed GO cells and were subsequently recycled with i.p. injection. GFP-positive cells isolated from the resulting omental tumors were designated GO2 cells (Fig. 1a).

Omental tumor weight and the number of tumor nodules in the peritoneal cavity at 70 days were markedly increased in mice injected with GO2 cells than for those injected with ID8G cells (Fig. 1b, c). The volume of ascites fluid was also greater for mice injected with GO2 cells than for those injected with ID8G cells, although this difference was not statistically significant. In addition, mice injected with GO2 cells had significantly shorter median survival time compared with those injected with ID8G cells (70 versus 122 days, $P = 0.0011$) (Fig. 1d). These results thus indicate that the metastatic potential of GO2 cells is markedly greater than that of parental ID8G cells.

**Expression of the PEDF gene is associated with prognosis in OC.** To identify genes that give rise to the enhanced metastatic potential of GO2 cells, we examined differences in gene expression profiles between ID8G cells and GO2 cells. We identified 723 genes whose expression level was at least twice as high in GO2 cells than in ID8G cells (Fig. 1e, Supplementary Data 1). To evaluate the clinical relevance of the top 10 most upregulated genes in GO2 cells, we assessed their relation to patient survival in a cohort of 531 patients with high-grade serous OC in The Cancer Genome Atlas (TCGA). With the exception of the gene for PEDF (*SERPINF1*), the expression levels of these genes were not significantly associated with both overall survival (OS) and disease-free survival (DFS) (Supplementary Table 1 and 2). Higher abundance of PEDF mRNA and secretion of PEDF protein in GO2 cells compared with parental ID8G cells were confirmed (Fig.1f, g). Upregulation of PEDF was also observed in cancer cells isolated from the resulting ascites (Supplementary Fig. 1a, b). To rule out the possibility that elevation of PEDF expression in metastatic OC cells are a mouse-specific phenomenon, using SKOV3ip1 human OC cells, we generated SKOV3ip1-Om2 cells, which corresponded to mouse GO2 cells (Supplementary Fig. 1c). The abundance of PEDF mRNA and secretion of PEDF protein were higher in SKOV3ip1-Om2 cells than in parental SKOV3ip1 cells, which is similar to the observations between ID8G and GO2 cells (Supplementary Fig. 1d, e). Higher *SERPINF1* expression was associated with a shorter median OS (43.3 versus 48.7 months, $P = 0.005$) (Fig. 1h) and shorter median DFS (17.2 versus 20.2 months, $P = 0.007$) (Fig. 1i). We also found that *SERPINF1* expression was significantly higher in stage III/IV disease than in stage I/II disease (Fig. 1j), consistent with the notion that PEDF promotes metastatic disease. We further assessed whether *SERPINF1* expression was higher in metastatic tumors than in primary tumors using an RNA-sequencing dataset (GSE137237) which are consisted of 11 matched pairs of tumors. We found significantly higher PEDF mRNA expression in metastatic tumors in the peritoneal cavity than in primary tumors (Fig. 1k). Together, these data confirm the dissemination-promoting function of PEDF not only in mice, but also in human ovarian cancer. We therefore subsequently focused on the role of PEDF in intraperitoneal dissemination.

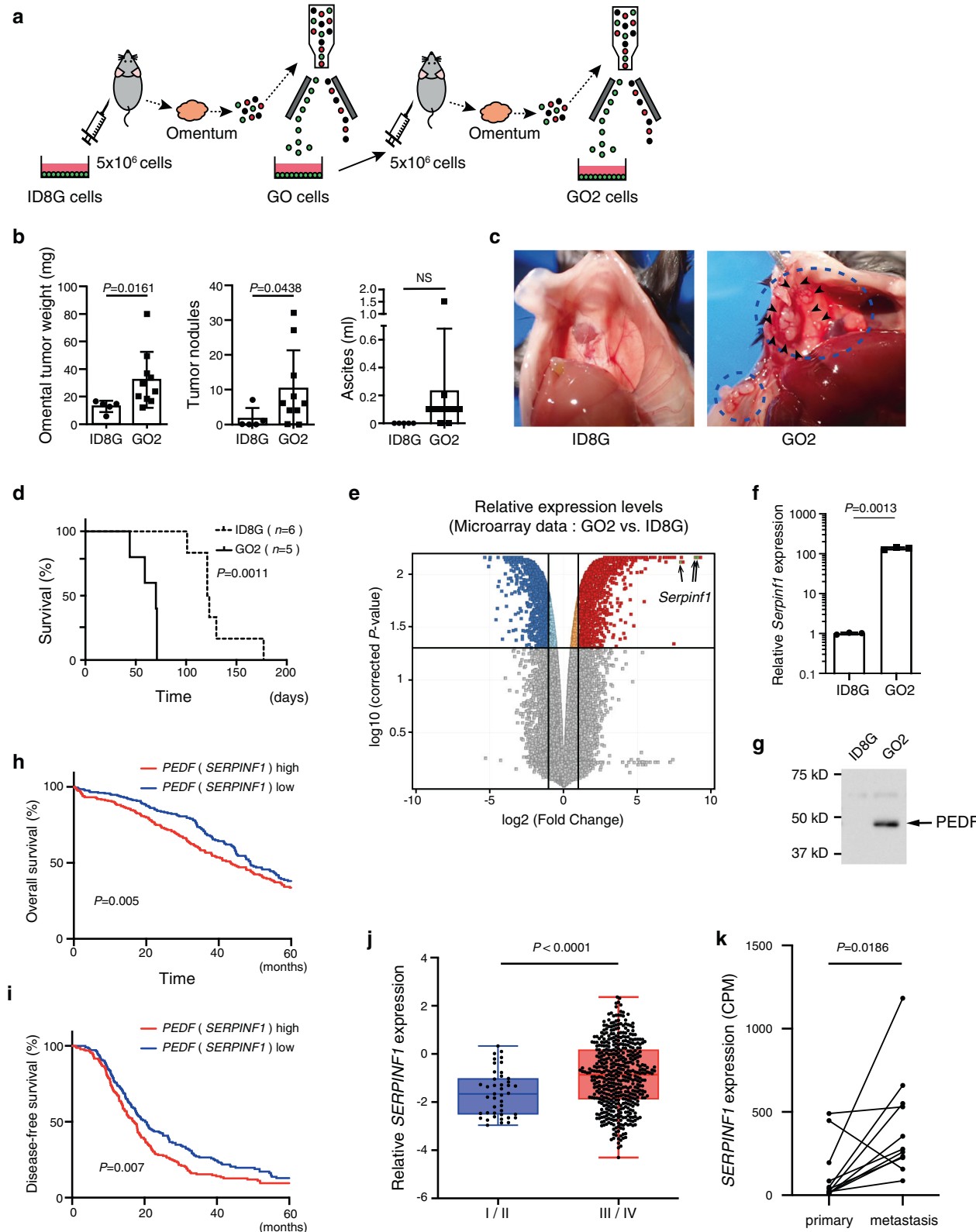

**PEDF promotes peritoneal dissemination of both mouse and human OC cells.** To clarify the role of PEDF expression in intraperitoneal dissemination of OC, we stably infected ID8G cells with a retrovirus that encodes mouse PEDF (ID8G-PEDF cells) or with corresponding empty vector (ID8G-EV cells). The abundance of PEDF mRNA and secretion of PEDF protein were increased in ID8G-PEDF cells compared with ID8G-EV cells

(Supplementary Fig. 2a, b). At 90 days after cell injection, recipient mice that received ID8G-PEDF cells exhibited significantly greater omental tumor weight, number of disseminated tumor nodules, and amount of ascites fluid than those that received ID8G-EV cells (Fig. 2a). Survival time was also significantly decreased in mice injected with ID8G-PEDF cells (96 versus 77 days, $P = 0.02$) (Fig. 2b). To exclude the possibility of artifacts

**Fig. 1 Pigment epithelium-derived factor (PEDF) is closely associated with increased intraperitoneal dissemination and prognosis in OC. a** Generation of a mouse ovarian tumor model based on the injection of GFP[+] omental tumor cells into the peritoneal cavity. **b** Omental tumor weight, the number of tumor nodules, and the volume of ascites fluid in C57BL/6 J mice at 70 days after i.p. injection of ID8G ($n = 5$ mice) or GO2 ($n = 10$ mice) cells. Data are means ± SD. **c** Representative macroscopic appearance of the peritoneum at 70 days after ID8G or GO2 cell injection. The blue dashed circle and black arrowheads indicate disseminated tumors. **d** Survival curves for mice injected with ID8G or GO2 cells ($n = 5$ each). The *P* value was determined with the log-rank test. **e** Volcano plot of the microarray data representing differentially expressed genes for GO2 cells versus ID8G cells. The data for all genes is plotted as log2 fold change versus the −log10 of the corrected *p* value. Thresholds are shown as lines on X-axis and Y-axis (fold change = 2.0, *P* value = 0.05, respectively). **f** RT-qPCR analysis of PEDF mRNA in ID8G and GO2 cells. **g** Immunoblot analysis of PEDF in culture supernatants of ID8G and GO2 cells. **h, i** Kaplan–Meier analysis of overall survival (**h**) and disease-free survival (**i**) for 529 patients with ovarian serous adenocarcinoma and high or low expression of the PEDF gene (*SERPINF1*) in TCGA. The expression cutoff was the median, and the *P* values were determined with the Wilcoxon test. **j** *SERPINF1* expression in tumor specimens of patients with early (I/II)–stage or late (III/IV)–stage OC in TCGA ($n = 534$). Data are presented as box-and-whisker plots, with the boxes indicating the median and quartile values and the bars indicating the range. **k** *SERPINF1* expression in 11 matched pairs of primary and metastatic human ovarian cancer samples based on the RNA-sequencing dataset[60]. Data were analyzed by Wilcoxon matched pairs signed rank test. Data for **b**, **f**, **j** were analyzed by unpaired *t* test with Welch's correction.

associated with retroviral infection, we generated 27 single-cell clones from ID8G cells. The clonal subpopulations varied in PEDF expression level, and three PEDF[high] clones (clones 1, 2, and 3) and two PEDF[low] clones (clones 26 and 27) were selected for subsequent analysis (Supplementary Fig. 2c). Omental tumor weight, number of disseminated tumor nodules, and volume of ascites fluid were greater for mice injected with the PEDF[high] clones than for those injected with the PEDF[low] clones (Fig. 2c). Survival time was also shorter for mice injected with PEDF[high] clones (Fig. 2d). Expression levels of PEDF and survival time of mice injected with corresponding cells showed significantly negative correlation (Fig. 2e). Together, these results suggest that PEDF[high] cells are largely responsible for cell colonization and expansion into the peritoneal cavity.

To examine the effect of PEDF expression on human tumor cell dissemination, we performed similar experiments with nude mice injected with SKOV3ip1 human OC cells stably over-expressing human PEDF (SKOV3ip1-PEDF cells). The abundance of PEDF mRNA and secretion of PEDF protein were increased in SKOV3ip1-PEDF cells compared with SKOV3ip1-EV cells (Supplementary Fig. 2d, e). Expression of human PEDF in the transplanted OC cells also promoted peritoneal dissemination and shortened the survival of the recipient mice (41.5 versus 36.5 days, $P = 0.0182$) (Fig. 2f, g).

**Marginal effect of PEDF on OC cells in vitro**. The formation of metastatic lesions by OC cells comprises four key steps: (1) exfoliation from the primary tumor site, (2) development of resistance to anoikis in peritoneal fluid, (3) attachment to peritoneal organs, and (4) colonization at metastatic sites. The cancer cells in our i.p. injection models can be considered to be the equivalent of exfoliated cells. We therefore next assessed the possible effects of PEDF on anoikis resistance, adhesion to an extracellular matrix protein, and cell proliferation in OC cells in vitro. Forced expression of PEDF did not affect anoikis in either ID8G or SKOV3ip1 cells (Fig. 3a). Given that fibronectin produced by mesothelial cells has been shown to influence peritoneal OC metastasis[23], we examined the attachment of OC cells to fibronectin. Expression of PEDF was found to have no effect on the adhesion of either ID8G or SKOV3ip1 cells to fibronectin (Fig. 3b). Cell proliferation was slightly inhibited in SKO-V3ip1 cells but was unaffected in ID8G cells by forced expression of PEDF (Fig. 3c, d).

To investigate further the function of PEDF in OC cells in vitro, we depleted GO2 cells of the protein through the stable introduction of one of two independent shRNA vectors. Knock-down of PEDF in the resulting GO2 sh#1 and sh#2 cells was confirmed by reverse transcription (RT) and quantitative PCR (qPCR) analysis, and additionally by immunoblot analysis

(Supplementary Fig. 3a, b). The shRNA-mediated depletion of PEDF had no effect on anoikis resistance or adhesion to fibronectin (Supplementary Fig. 3c, d), but had a slight inhibitory effect on cell proliferation (Supplementary Fig. 3e).

Association between peritoneal metastasis, survival, and poor chemoresponse and epithelial-mesenchymal transition (EMT) have been reported in OC patients[24,25]. Therefore, we assessed the effects of PEDF on EMT-related proteins, E-cadherin and vimentin. Expression of PEDF was not associated with E-cadherin or Vimentin expression (Supplementary Fig. 3f) Together, these results thus indicate that PEDF has only marginal effect on OC cells in vitro. Therefore, we hypothesized that PEDF promotes dissemination via some in vivo-specific mechanism.

**Effect of PEDF on OC cells in vivo**. The omentum is considered to be an essential site for cancer cell seeding and contributes to peritoneal dissemination in ovarian cancer[26,27]. To determine whether PEDF promotes dissemination via the omentum, we assessed the dissemination potency of ID8G-PEDF cells using omentectomized mice (Fig. 3e). Although the number of tumor nodules in the peritoneal cavity and the amount of ascites tended to be lower in omentectomized mice than in sham-operated mice, the differences were not significant (Fig. 3f). This finding suggests that the contribution of not only omentum but also additional factors such as peritoneal fluid to the PEDF-mediated peritoneal dissemination.

We next assessed the survival of ID8G-PEDF and ID8G-EV cells after their injection into the peritoneal cavity (Fig. 3g). The number of surviving OC cells was greater for ID8G-PEDF cells than for ID8G-EV cells on day 5 and 8, with the number of both cells decreasing gradually after injection (Fig. 3h). Conversely, the number of surviving GO2 sh#1 or sh#2 cells after i.p. injection was greatly reduced compared with that for corresponding control cells (Fig. 3i). Collectively, these data indicate that PEDF facilitates the escape of OC cells from immune surveillance at an early stage of dissemination.

**Macrophages play a critical role in OC cell survival in the peritoneal cavity**. Ovarian cancer is characterized by a unique peritoneal tumor microenvironment enriched with abundant immune cells, including macrophages and T cells[28]. A previous study using the ID8 model showed no significant difference in the tumor burden between immunocompetent mice and mice lacking mature T and B cells[29]. In contrast, the depletion of macrophages promoted tumor growth and shortened the survival time independently of T and B cells. Furthermore, macrophages are reported to be essential in spheroid formation of OC cells in the peritoneal cavity and contribute to the dissemination of OC cells

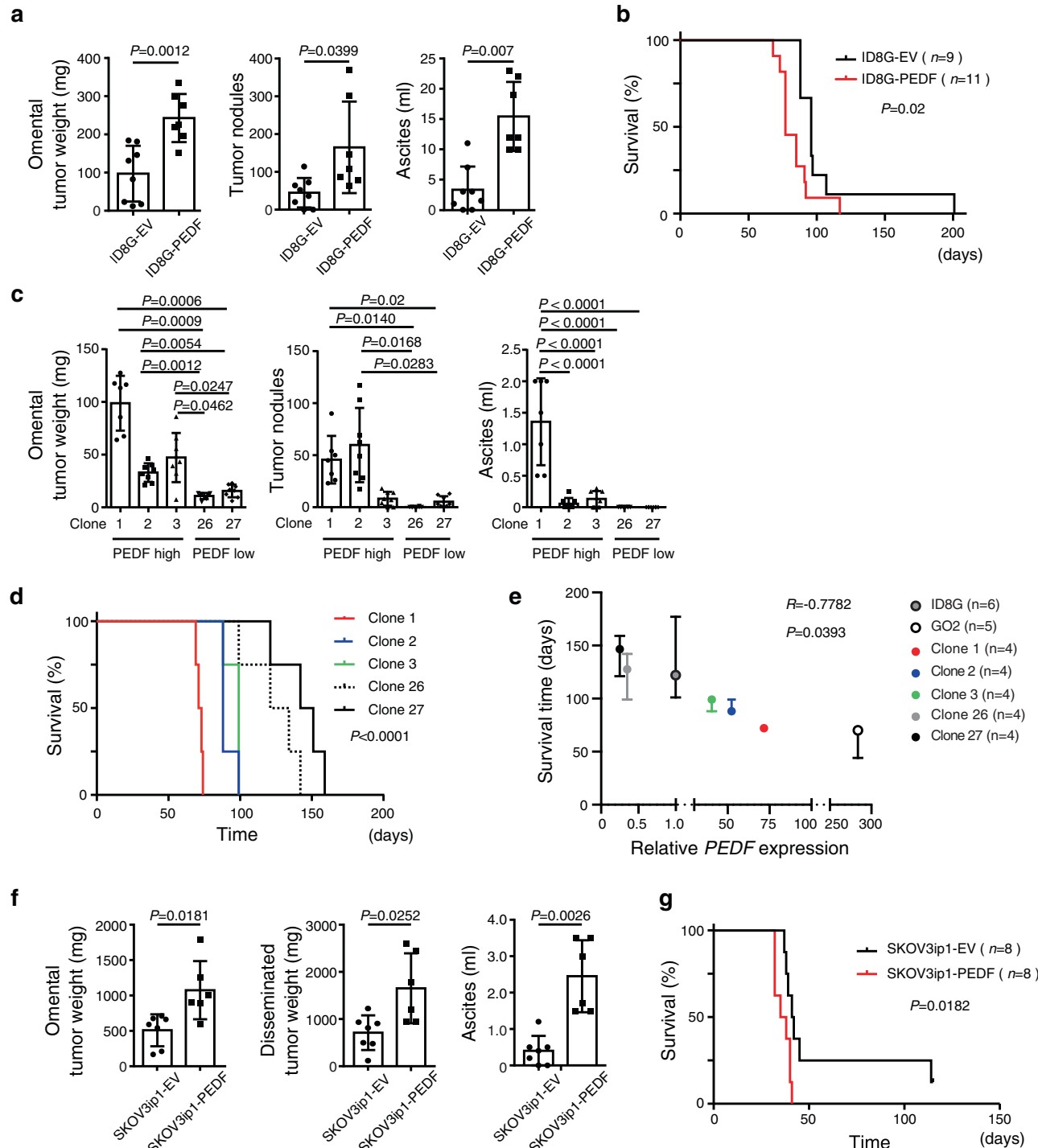

**Fig. 2 PEDF expression is associated with increased intraperitoneal dissemination of both mouse and human OC cells and with poor prognosis.**
**a** Omental tumor weight, the number of tumor nodules, and the volume of ascites fluid for C57BL/6 J mice at 90 days after injection of ID8G-EV ($n = 8$ mice) or ID8G-PEDF ($n = 7$ mice) cells into the peritoneal cavity. Data are means ± SD. **b** Survival curves for mice injected with ID8G-EV ($n = 9$) or ID8G-PEDF cells ($n = 11$). The P value was determined with the log-rank test. **c** Omental tumor weight, number of tumor nodules, and the volume of ascites fluid for C57BL/6 J mice at 63 days after injection of PEDF^high or PEDF^low clones of ID8G cells into the peritoneal cavity. Data are means ± SD ($n = 7$ or 8 mice). Data for omental tumor weight and number of tumor nodules were analyzed by Welch's ANOVA followed by Dunnett's post hoc test and for ascites by Tukey's multiple comparisons test. **d** Survival curves for mice injected with PEDF^high or PEDF^low clones of ID8G cells ($n = 4$ each). The P value was determined with the log-rank test. **e** Pearson's correlation analysis for expression of PEDF mRNA versus survival time of mice inoculated with the corresponding cells. Data are median ± 95% CI. **f** Omental and intraperitoneal disseminated tumor weight as well as the volume of ascites fluid in nude mice at 27 days after injection SKOV3ip1-EV cells ($n = 7$ mice) or SKOV3ip1-PEDF cells ($n = 6$ mice) into the peritoneal cavity. Data are means ± SD. **g** Survival curves for mice injected with SKOV3ip1-EV or SKOV3ip1-PEDF cells ($n = 8$ each). The P value was determined with the log-rank test. Data for **a** and **f** were analyzed by unpaired t test with Welch's correction.

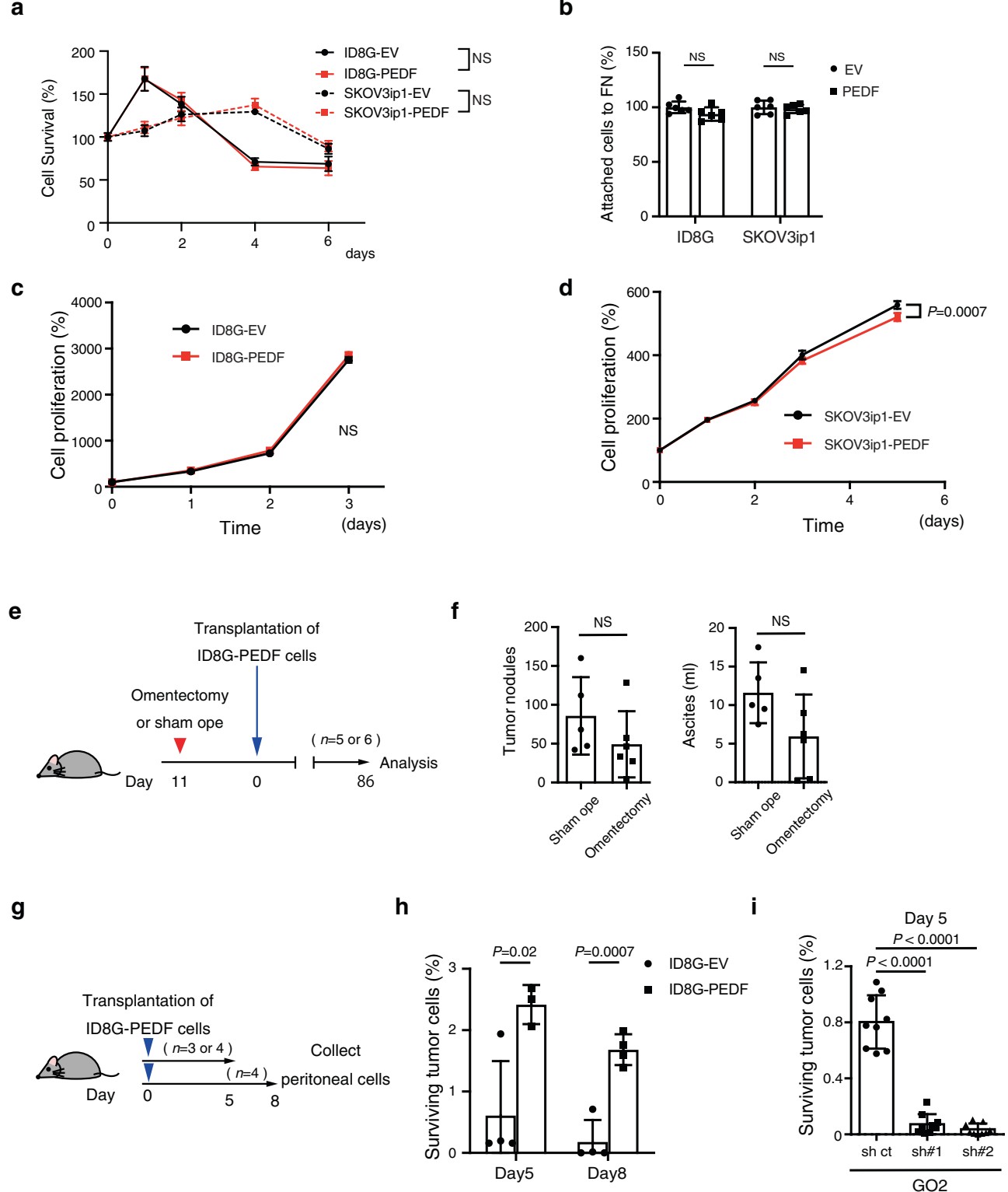

in the ID8 model[30]. These reports indicate the prominent role of macrophages in OC cell dissemination.

We found that SKOV3ip1-PEDF could promote peritoneal dissemination in Balb/c-nu/nu mice (Fig. 2f, g), which are unable to produce T cells; this would suggest that PEDF functions independently of T cells. To further confirm this, we assessed the peritoneal dissemination of ID8G-PEDF cells compared to ID8G-EV cells in Balb/c-nu/nu mice. ID8G-PEDF cells disseminated more robustly into the peritoneal cavity than did ID8G-EV cells,

as indicated by greater omental tumor weight, number of peritoneal tumor nodules, and amount of ascites at Day 64 post injection (Fig. 4a). Mice in the ID8G-PEDF group also had decreased survival time than the ID8G-EV group (125.5 versus 103.0 days, $P = 0.0001$) (Fig. 4b). These results indicate that PEDF promotes peritoneal dissemination independently of T cells.

We showed that PEDF plays a critical role in OC cell survival in the peritoneal cavity at the early stage of dissemination

**Fig. 3 Effects of PEDF on OC cells. a** Anoikis resistance assay for ID8G-EV and ID8G-PEDF cells ($n = 5$ each) and for SKOV3ip1-EV and SKOV3ip1-PEDF cells ($n = 6$ each). **b** Attachment assay for ID8G-EV and ID8G-PEDF cells and for SKOV3ip1-EV and SKOV3ip1-PEDF cells cultured in wells coated with fibronectin ($n = 6$ each). **c, d** Cell proliferation assay for ID8G-EV and ID8G-PEDF cells (**c**) ($n = 5$ each) and for SKOV3ip1-EV and SKOV3ip1-PEDF cells (**d**) ($n = 6$ each). All data are means ± SD for the indicated number of replicates from representative experiments out of a total of three performed. **e** Omentectomized or sham-operated C57BL/6 J mice were injected i.p. with ID8G-PEDF cells into the peritoneal cavity. **f** The number of tumor nodules and the volume of ascites fluid for omentectomized ($n = 6$ mice) or sham-operated ($n = 5$ mice) C57BL/6 J mice at 86 days after injection of ID8G-PEDF cells into the peritoneal cavity. Data are means ± SD. **g** C57BL/6 J mice were injected i.p. with ID8G-EV or ID8G-PEDF cells into the peritoneal cavity. At each time point, mice were sacrificed and peritoneal cells were collected for further analysis. **h** Number of GFP+ cells in peritoneal washes at the indicated times after i.p. injection of ID8G-EV or ID8G-PEDF cells ($5 \times 10^6$) into C57BL/6 J mice ($n = 3$ or 4 mice for each group at each time point as in **g**). **i** Surviving GFP+ tumor cells in the peritoneal cavity of C57BL/6 J mice at 5 days after i.p. injection of GO2 sh#1 or sh#2 or corresponding control (sh ct) cells ($n = 9$ mice per group). All data are means ± SD. Data for **a**, **c**, **d** were analyzed by two-way ANOVA. Data for **f**, **h**, **i** were analyzed by unpaired $t$ test with Welch's correction.

(Fig.3g–i). Next, to clarify the relation between PEDF expression in OC cells and macrophages at the early stage of dissemination, we examined the effect of macrophage depletion by intravenous injection of liposomal clodronate before transplantation of ID8G-PEDF cells (Fig. 4c). Liposomal clodronate itself did not affect cell survival of either ID8G-EV or ID8G-PEDF cells (Supplementary Fig. 3g). The depletion of macrophages resulted in significant attenuation of peritoneal tumor cell survival on day 5, suggesting the supportive role of macrophages on tumor cell survival in the peritoneal cavity at the early stage of dissemination (Fig. 4d).

To further characterize the mechanisms by which PEDF promotes tumor cell escape from immune surveillance, we collected cells from the peritoneal cavity of recipient mice at 5 days after transplantation of ID8G-EV or ID8G-PEDF cells and analyzed them for immune cell markers by flow cytometry. The number of CD45+ cells was significantly decreased in mice injected with ID8G-PEDF cells than in those injected with ID8G-EV cells. Furthermore, injection of ID8G-PEDF cells induced a significant decrease in the number of CD11b+ myeloid cells but did not affect other immune cell types including T cells, B cells, NK cells, and dendritic cells (Fig. 4e). These results provide further support for the hypothesis that PEDF promotes peritoneal dissemination of OC not through T cells, but through the function of myeloid cells.

**PEDF induces CD206+ IL-10-producing macrophages.** Peritoneal macrophages mainly consist of two distinct subsets. One subset is F4/80+ MHC II− large peritoneal macrophages (LPMs), which are most abundant, and the other is a minor population of F4/80− MHC II+ small peritoneal macrophages (SPMs)[31]. Additionally, a previous study showed that the inoculation of cancer cells induced gradual replacement of LPMs with F4/80int MHC IIint macrophages (intPMs)[32]. Therefore, to clarify which subset is the major macrophage population in our experiments, we assessed the expression of F4/80 and MHC II in peritoneal CD11b+ cells. F4/80+ MHC II− LPMs were the major population and represented approximately 70% of peritoneal macrophages both in mice inoculated with ID8G-EV cells and ID8G-PEDF cells (Fig. 5a, Supplementary Fig. 4a). Macrophages in the tumor microenvironment have been found to be highly plastic cells and can adopt either antitumor or protumor features[33,34]. We therefore further investigated the phenotype of intraperitoneal macrophages at 5 days after i.p. transplantation of ID8G-EV or ID8G-PEDF cells. PEDF expression in the transplanted cells was associated with an increase in the percentage of CD206+ cells in F4/80+ to int macrophages (LPMs and intPMs) but not with F4/80− SPMs (Fig. 5b, Supplementary Fig. 4b).

CD206+ macrophages produce several cytokines and chemokines including IL-10, TGF-β, and CCL22, with IL-10 being a key immunosuppressive cytokine[33,35]. To evaluate the effects of PEDF on IL-10 production in peritoneal macrophages, we collected peritoneal macrophages from mice by MACS and analyzed their IL-10 production. Injection of ID8G-PEDF cells caused significantly higher IL-10 production in peritoneal macrophages than that with ID8G-EV cells (Fig. 5c). Moreover, IL-10 concentration in peritoneal wash fluid at 5 days after OC cell transplantation was significantly higher for mice injected with ID8G-PEDF cells than for those injected with ID8G-EV cells (Fig. 5d).

To further assess the effects of PEDF on macrophages, we isolated CD45+CD11b+F4/80+ macrophages from the peritoneal cavity of mice by FACS 2 days after OC cell transplantation. RT-qPCR analysis of the isolated cells revealed that the abundance of IL-10 and CD206 mRNAs was significantly higher in mice injected with ID8G-PEDF cells than in mice injected with ID8G-EV cells (Fig. 5e). We also examined the possible effects of PEDF on macrophages in vitro with the use of bone marrow-derived macrophages (BMDMs) or RAW 264 murine macrophage cells. Culture supernatant of ID8G-PEDF cells increased the amounts of IL-10 and CD206 mRNAs in BMDMs and RAW 264 cells (Fig. 5f, g).

**Blocking IL-10 signaling suppresses OC cell survival in the peritoneal cavity.** To determine whether the PEDF–IL-10 axis promotes intraperitoneal OC dissemination, we injected ID8G-EV cells or ID8G-PEDF cells into the peritoneal cavity of C57BL/6 J mice after the onset of treatment with either antibodies to the IL-10 receptor (IL-10R) or the IL-10 inhibitor AS101 (Fig. 5h). Treatment with either anti–IL-10R or AS101 resulted in a significant reduction in ID8G-PEDF cell survival in the peritoneal cavity, while no significant effect was observed in ID8G-EV cells (Fig. 5i). These findings suggest that induction of immunosuppressive macrophages by PEDF supports OC cell survival in the peritoneal cavity.

**Serum and ascites PEDF are elevated in OC patients and correlate with early recurrence.** On the basis of our data obtained from mice, we examined the relation between the expression of the PEDF gene (SERPINF1) and that of genes for macrophage markers in tumor specimens of OC patients in TCGA. Expression of SERPINF1 was correlated with that of the immunosuppressive macrophage marker genes IL10, CD163, MSR1, and TGFB1 but not with that of the proinflammatory macrophage markers IL12A and NOS2 (Fig. 6a). These results thus suggest PEDF also induce immunosuppressive macrophages in human OCs.

As PEDF is secretory protein, we next investigated whether PEDF in body fluid could serve as clinical biomarker in OC. We measured PEDF levels in ascites specimens from women with serous OC ($n = 28$) and from those with low malignancy potential (LMP) ovarian tumors ($n = 3$). The PEDF concentration in ascites fluid was significantly higher for the patients with OC than for those with LMP tumors (Fig. 6b). To determine

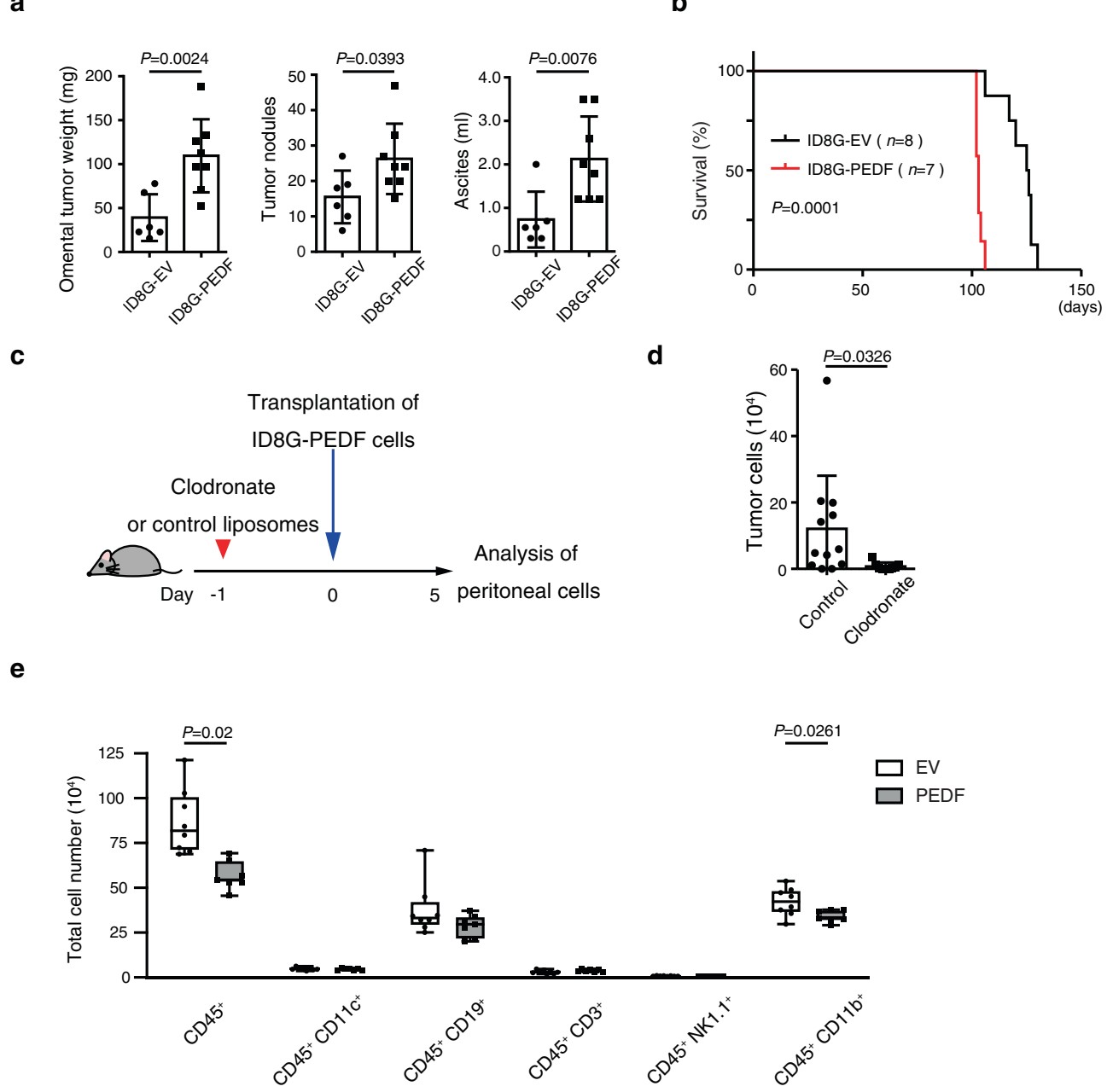

**Fig. 4 Macrophages play a critical role in OC cell survival in the peritoneal cavity. a** Omental tumor weight, the number of tumor nodules, and the volume of ascites fluid at 64 days after injection of ID8G-EV ($n = 6$ mice) or ID8G-PEDF ($n = 8$ mice) cells into the peritoneal cavity of BALB/c-nu/nu mice. Data are means ± SD. **b** Survival curves for BALB/c-nu/nu mice injected with ID8G-EV ($n = 8$) or ID8G-PEDF ($n = 7$) cells. The $P$ value was determined with the log-rank test. **c** C57BL/6 J mice were injected i.v. with control or clodronate liposomes 1 day before injection of ID8G-PEDF cells into the peritoneal cavity. **d** Flow cytometric analysis of the numbers of GFP+ tumor cells in the peritoneal cavity of mice at 5 days after injection of ID8G-PEDF cells in mice treated as in **c**. Data are means ± SD ($n = 12$ or 9 mice for control and clodronate liposomes, respectively). **e** Flow cytometry quantified CD45+ cells, CD45+ CD11c+ dendritic cells, CD45+ CD19+ B cells, CD45+ CD3+ T cells, CD45+ CD3− NK1.1+ NK cells, and CD45+ CD11b+ myeloid cells in the peritoneal cavity, 5 days after i.p. injection with ID8G-EV ($n = 8$) or ID8G-PEDF cells ($n = 7$) into C57BL/6 J mice. Data are presented as box-and-whisker plots. Data for **a**, **d**, **e** were analyzed by unpaired $t$ test with Welch's correction.

whether ascites PEDF predict immunosuppressive microenvironment in the peritoneal cavity, we next examined the relation between ascites PEDF and ascites IL-10. IL-10 concentration was higher in OC patients than in patients with LMP tumors (Fig. 6c). We also found that IL-10 and PEDF concentrations were positively correlated in ascites samples (Fig. 6d). Because higher ascites IL-10 levels were reported to be associated with poor prognosis[36], we next analyzed relationships between ascites PEDF

and clinicopathological factors. As shown in Table 1, high ascitic PEDF levels had a trend toward more recurrences within 2 years.

Given that the PEDF level in serum would provide a less invasive, less expensive, and more convenient repeatable biomarker than that in ascites, we investigated whether serum PEDF at diagnosis could be substituted for ascites PEDF. Serum PEDF concentration was significantly higher for the patients with OC than for those with benign or LMP tumors, which is

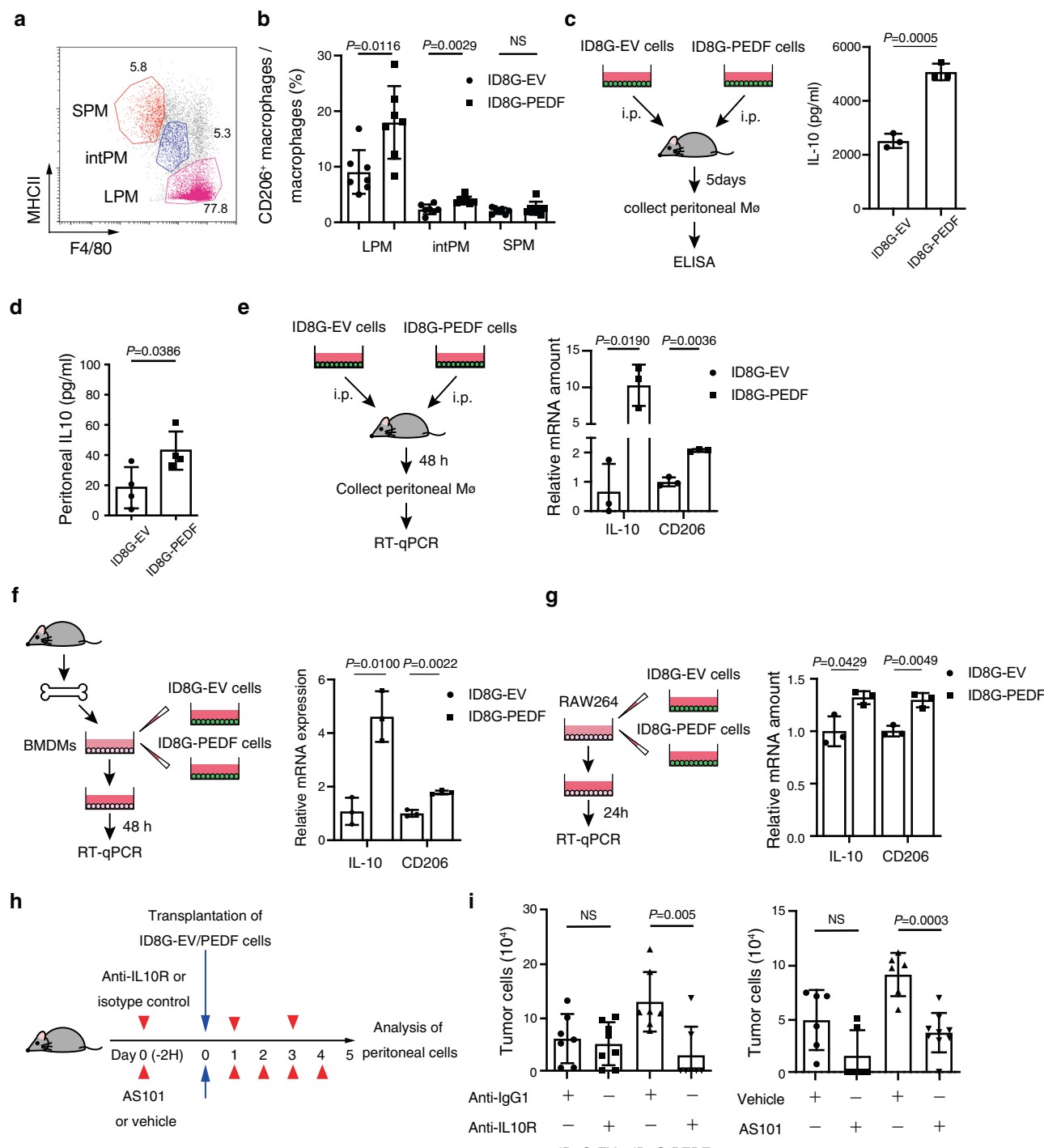

**Fig. 5 PEDF-induced immunosuppressive macrophages contribute to tumor cell survival in the mouse peritoneal cavity. a** Flow cytometric analysis of peritoneal myeloid cells (CD45+CD11b+ cells) 5 days after intraperitoneal injection of OC cells. LPMs were subsequently gated as F4/80+ MHC II− (pink), intPM as F4/80int MHC IIint (blue), and SPMs as F4/80- MHC II+(red). **b** Percentage of CD206+ macrophages among LPMs, intPMs, and SPMs in the peritoneal cavity of C57BL/6 J mice at 5 days after i.p injection of ID8G-EV or ID8G-PEDF cells (*n* = 7 mice). **c** IL-10 ELISA from peritoneal macrophages of C57BL/6 J mice injected with ID8G-EV or ID8G-PEDF cells (*n* = 3). **d** IL-10 concentration in peritoneal wash fluid collected at 5 days after OC cell transplantation (*n* = 4 mice each). **e** RT-qPCR analysis of IL-10 and CD206 mRNAs in peritoneal CD45+CD11b+F4/80+ macrophages (Mø) isolated at 48 h after i.p. injection of C57BL/6 J mice with ID8G-EV or ID8G-PEDF cells (*n* = 3 mice each). **f** RT-qPCR analysis of IL-10 and CD206 mRNAs in BMDMs isolated from C57BL/6 J mice and treated with conditioned medium of ID8G-EV or ID8G-PEDF cells for 48 h (*n* = 3 replicates from one of two independent experiments). **g** RT-qPCR analysis of IL-10 and CD206 mRNAs in RAW 264 cells treated with conditioned medium of ID8G-EV or ID8G-PEDF cells for 24 h (*n* = 3 replicates from one of two independent experiments). **h** C57BL/6 J mice were injected i.p. with ID8G-EV or ID8G-PEDF cells after the onset of repeated i.v. administration of antibodies to IL-10R (or control IgG) or AS101 (or vehicle). **i** Flow cytometric analysis of the number of GFP+ (tumor) cells in the peritoneal cavity at 5 days after ID8G-EV or ID8G-PEDF cell injection in mice treated with anti–IL-10R (*n* = 7 or 8) or control IgG (*n* = 7 each) or with AS101 (*n* = 6 or 9) or vehicle (*n* = 6 each) as in **h**. All data are means ± SD and were analyzed by unpaired *t* test with Welch's correction.

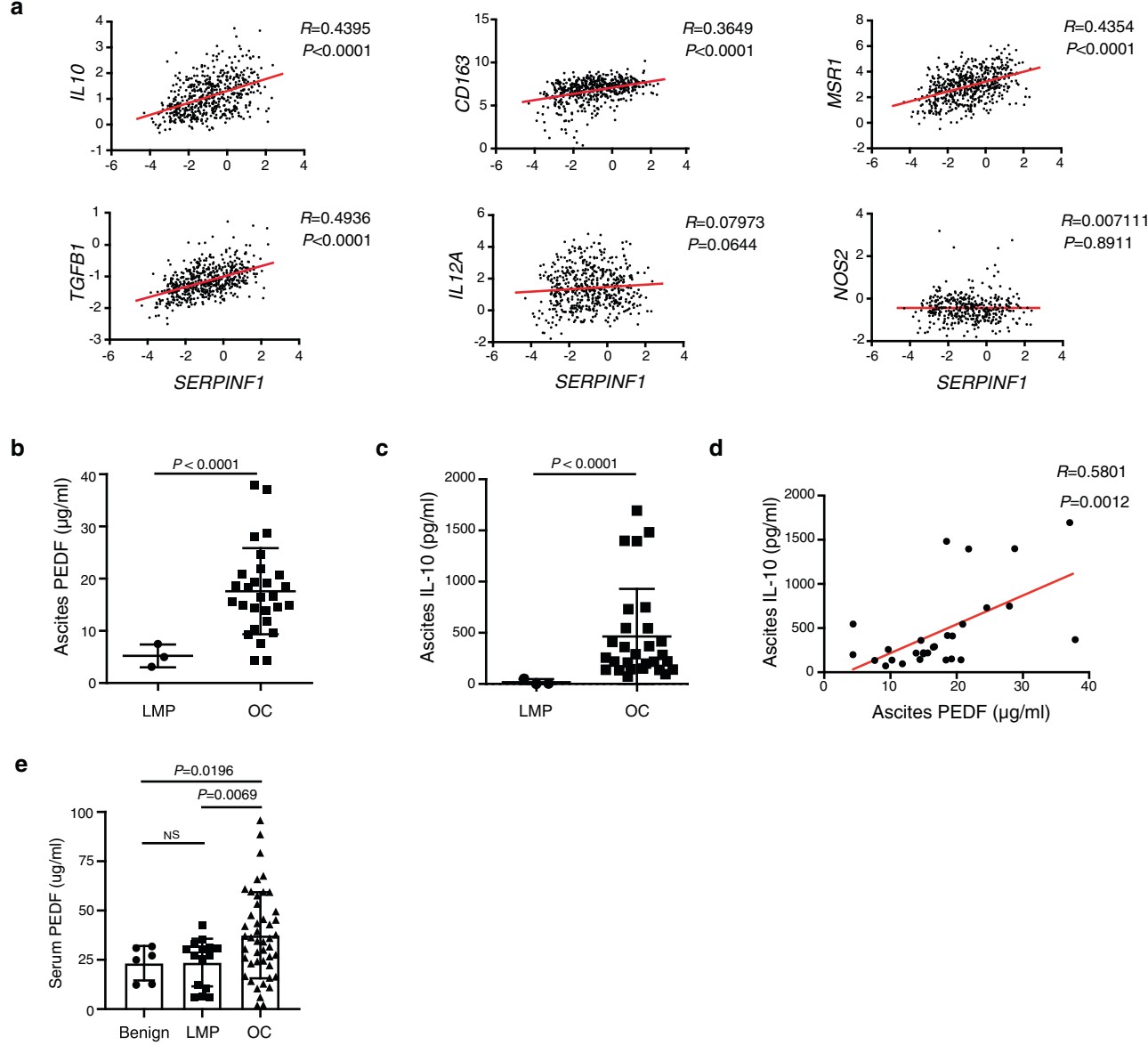

**Fig. 6 PEDF levels in ascites fluid and serum samples are increased in OC patients. a** Pearson's correlation analysis for expression of *SERPINF1* versus that of *IL10*, *CD163*, *MSR1*, *TGFB1*, *IL12A*, or *NOS2* in human OC samples of TCGA ($n = 531$). **b** PEDF concentration as measured with an ELISA in ascites specimens of patients with LMP ovarian tumors ($n = 3$) or OC ($n = 28$). **c** Concentration of IL-10 in ascites samples of patients with LMP tumors ($n = 3$) or OC ($n = 28$) as measured with an ELISA. **d** Pearson's correlation analysis for IL-10 and PEDF concentrations in ascites specimens of patients with OC ($n = 28$). **e** Serum PEDF concentration of patients with benign ovarian tumors ($n = 6$), LMP tumors ($n = 15$), or OC ($n = 46$) as measured with an ELISA. Data are means ± SD and were analyzed by Welch's ANOVA followed by Dunnett's post hoc test. Data for **b**, **c** are means ± SD and were analyzed by unpaired *t* test with Welch's correction.

consistent with our findings in ascites samples (Fig. 6e). Notably, high serum PEDF levels were significantly associated with a high recurrence rate within two years (Table 1). Together, these findings suggest that PEDF could be a biomarker for diagnosis of OCs and predict recurrence in OC patients.

**Histone acetylation contributes to regulation of PEDF gene expression.** We next investigated the regulation of PEDF gene expression. Given that the ID8 cell line was generated by spontaneous transformation of mouse ovarian surface epithelial cells (MOSECs), we studied two normal MOSEC lines (p53-def-MOSE and T-Ag-MOSE) for comparison. Immunoblot analysis revealed that the amount of secreted PEDF was substantially lower for ID8G and GO cells than for normal MOSEC lines and GO2 cells

(Fig. 7a). Given that protein abundance is controlled through multiple mechanisms including those that operate at the levels of transcription, translation, and proteolysis, we next examined the amount of PEDF mRNA in these various cells. Similar to protein expression, the amount of PEDF mRNA was lower in ID8G and GO cells than in normal MOSECs and GO2 cells (Fig. 7b), suggesting that PEDF expression is regulated at the transcriptional or epigenetic level in association with OC initiation and progression.

To determine whether histone acetylation or demethylation correlates with activation of PEDF gene expression, we treated p53-def-MOSE and GO2 cells with histone acetyltransferase (HAT) inhibitors or histone demethylase inhibitors. Treatment of both cell lines with HAT inhibitors significantly suppressed expression of the PEDF gene (Fig. 7c), whereas histone

**Table 1 Clinical and histopathologic characteristics of the study population.**

| Clinical characteristics | Ascites PEDF (µg/ml) | | | Serum PEDF (µg/ml) | | |
|---|---|---|---|---|---|---|
| | <16.6 ($n = 14$) | >16.6 ($n = 14$) | P | <16.6 ($n = 9$) | >16.6 ($n = 14$) | P |
| Age, mean ± SD | 61.6 ± 9.93 | 63.86 ± 8.96 | 0.54 | 60.11 ± 11.8 | 63.57 ± 10.1 | 0.50 |
| FIGO stage | | | 0.48 | | | 0.54 |
| I/II | 0 | 2 (14) | | 2 (22) | 1 (7) | |
| III/IV | 14 (100) | 12 (86) | | 7 (78) | 13 (93) | |
| LN metastasis | | | >0.99 | | | >0.99 |
| Yes | 2 (14) | 2 (14) | | 1 (11) | 1 (7) | |
| No | 12 (86) | 12 (86) | | 8 (89) | 13 (93) | |
| Distant metastasis | | | 0.38 | | | 0.54 |
| Yes | 5 (36) | 2 (14) | | 2 (22) | 1 (7) | |
| No | 9 (64) | 12 (86) | | 7 (78) | 13 (93) | |
| Recurrence within 2 years(Stage II/III)[a] | | | 0.07 | | | 0.043 |
| Yes | 2 (22) | 7 (70) | | 1 (17) | 9 (75) | |
| No | 7 (78) | 3 (30) | | 5 (83) | 3 (25) | |

Values in the table are expressed as n (%).
FIGO International Federation of Gynecology and Obstetrics.
[a]Patients who achieved complete response after initial treatment were included.

demethylase inhibitors had no such effect (Fig. 7d), suggesting that PEDF gene expression is regulated at the level of histone acetylation.

Members of the bromodomain and extraterminal (BET) family of proteins bind to acetylated histones to promote gene transcription, and the gene locus for the BET-family protein BRD4 has been found to be frequently amplified in OC[37]. We therefore examined whether expression of *BRD4* and that of *SERPINF1* is correlated in human OC specimens, with the use of a published database containing gene expression profiles for tumor specimens obtained by laser capture and microdissection from 53 patients with high-grade serous OC[38]. Indeed, we detected a significant positive correlation between *BRD4* and *SERPINF1* expression in these specimens (Fig. 7e). We then examined the effect of BET inhibitors on PEDF gene expression. The small-molecule inhibitor JQ1, which competitively binds to the acetyl-lysine recognition pocket of the BET bromodomain, suppressed PEDF expression at both mRNA and protein levels and in a concentration-dependent manner in both p53-def-MOSE and GO2 cells (Fig. 7f, h). Another BET inhibitor, ARV-825, a hetero-bifunctional proteolysis-targeting chimera that induces degradation of BET-family proteins, similarly inhibited PEDF expression (Fig. 7g, h). Collectively, these results implicate histone acetylation and BET-family proteins in the regulation of PEDF gene expression.

**The BET inhibitor ARV-825 limits tumor cell survival and induction of CD206$^+$ macrophages in the peritoneal cavity.** JQ1 is a potent pan-BET inhibitor, with a broad target range limiting its specificity. Furthermore, JQ1 has a short half-life and its effective concentrations have been found to be higher than physiologically safe levels in vivo[39]. We, therefore, focused on inhibitor ARV-825 for in vivo experiments. We investigated the effect of BET inhibition on OC cell survival in vivo with the PEDF$^{high}$ GO2 cell line (Fig. 8a). ARV-825 treatment resulted in significant suppression of GO2 cell survival in the peritoneal cavity (Fig. 8b). Flow cytometric analysis of immune cell surface markers also revealed that ARV-825 treatment significantly increases the number of macrophages (CD45$^+$CD11b$^+$F4/80$^+$ cells) (Fig. 8c). However, the ratio of CD206+ macrophages to total macrophages is significantly decreased (Fig. 8d), suggesting the increase of CD206$^-$ macrophages which may promote the elimination of cancer cells. Together, these data supported the

notion that BET inhibition limits OC progression by depleting CD206$^+$ immunosuppressive macrophages.

**Discussion**

Here, we have shown that (1) PEDF promotes peritoneal dissemination of OC cells; (2) PEDF increases CD206$^+$ IL-10-producing macrophages in the peritoneal cavity; (3) the serum concentration of PEDF is elevated in OC patients and associated with high recurrence rate; and (4) BET-family protein inhibition attenuates the survival of OC cells through suppression of PEDF expression.

Forced expression of PEDF in OC cells promoted their peritoneal dissemination. Single-cell clone analysis revealed the heterogeneity of PEDF expression among OC cells, with PEDF$^{high}$ clones exhibiting higher dissemination capability compared with PEDF$^{low}$ clones. These results thus indicate that PEDF contributes to cancer cell dissemination, and that PEDF$^{high}$ cells selectively colonize the peritoneal cavity. PEDF has previously been shown to have an antimetastatic role by suppressing tumor cell invasion and migration in colon cancer and breast cancer[40–42], both of which metastasize via hematogenous and lymphatic routes. Given that extraperitoneal metastasis is relatively uncommon in OC, OC cells are thought to metastasize preferentially through direct dissemination instead of through a hematogenous route[43]. This difference in metastatic paths may account for the difference in PEDF function found in this current paper and previous studies. Our analysis of TCGA data revealed that higher PEDF gene expression was associated with shorter OS and DFS in women with OC. Furthermore, an analysis of 11 pairs of primary and metastatic tumors in patients with OC showed a significant increase in PEDF expression in metastatic tumors. On the other hand, the expression level of PEDF was found to be lower in ID8G OC cells than in normal MOSECs, suggesting that PEDF suppresses tumor initiation and growth. Together, these findings suggest that PEDF may play two opposing functional roles in OC: suppression of tumor initiation and promotion of tumor dissemination.

We found that PEDF increased the ratio of CD206$^+$ macrophages in the peritoneal cavity. A high CD206$^+$ to total macrophage ratio (indicative of an M2-like phenotype) in OC patients has previously been shown to be associated with shorter progression-free survival and OS[44], consistent with our data. In pancreatic cancer, PEDF was found to serve as an anti-inflammatory immunomodulator that inhibits macrophage

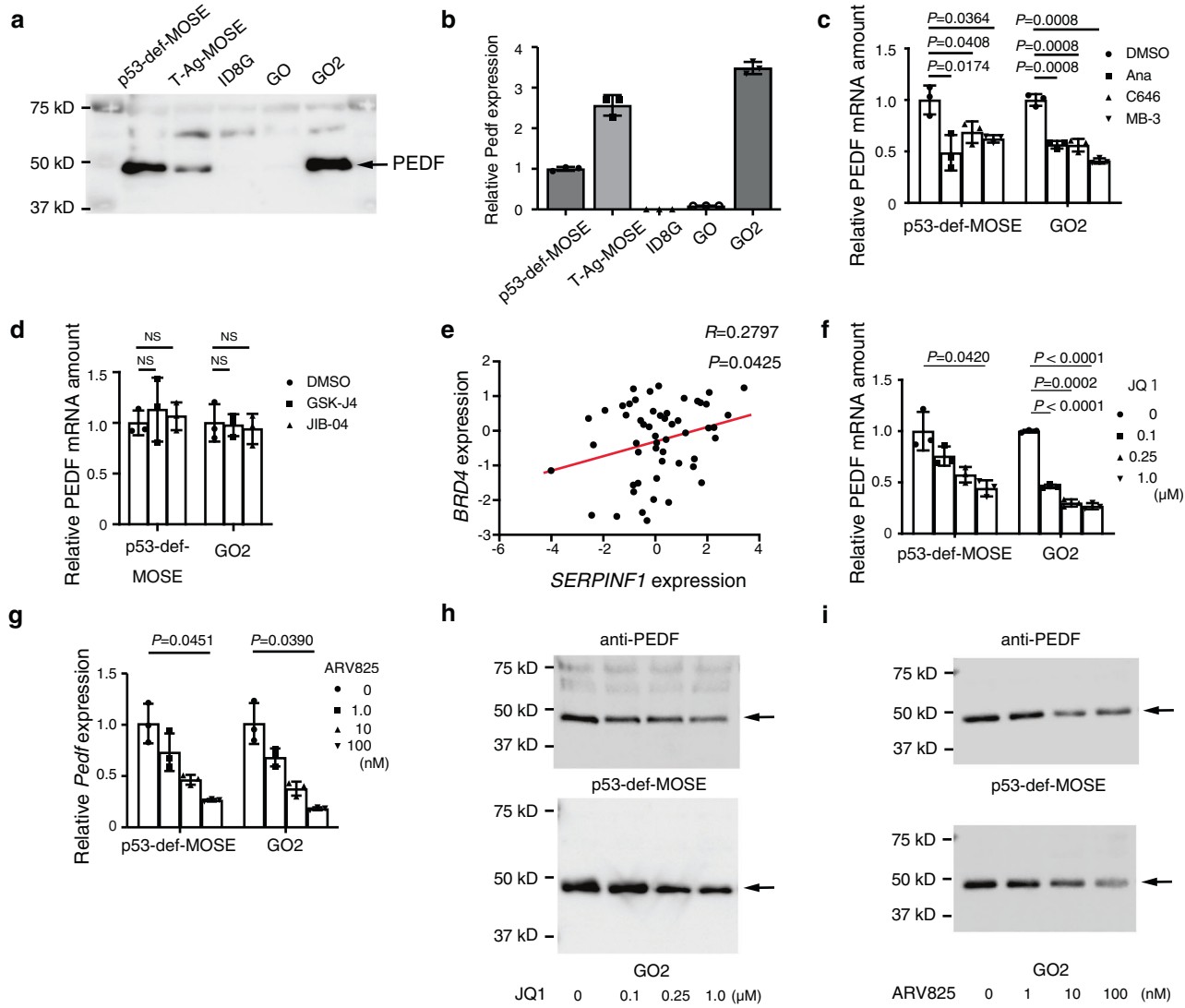

**Fig. 7 HAT inhibitors and BET inhibitors suppress PEDF expression in vitro. a** Immunoblot analysis of PEDF in culture supernatants of MOSEC, ID8G, GO, and GO2 cell lines. **b** RT-qPCR analysis of PEDF mRNA in cells as in **a. c**, **d** RT-qPCR analysis of PEDF mRNA in p53-def-MOSE and GO2 cells treated for 24 h with the HAT inhibitors anacardic acid (Ana, 10 μM), C646 (10 μM), or MB-3 (100 μM) (**c**) or with the histone demethylase inhibitors GSK-J4 (1.0 μM) or JIB-04 (250 nM) (**d**). Cells treated with DMSO were examined as a vehicle control. **e** Pearson's correlation analysis for *BRD4* and *SERPINF1* expression in 53 specimens of high-grade serous OC obtained by laser capture and microdissection[38]. **f**, **g** RT-qPCR analysis of PEDF mRNA in p53-def-MOSE and GO2 cells treated for 24 h with the BET inhibitors JQ1 (**f**) or ARV-825 (**g**) at the indicated concentrations. **h**, **i** Immunoblot analysis of PEDF in culture supernatants of cells treated as in **f** and **g**. All quantitative data are means ± SD of three replicates for representative experiments out of a total of three performed. Data for **c**, **d** were analyzed by unpaired *t* test with Welch's correction. Data for **f**, **g** were analyzed by Welch's ANOVA followed by Dunnett's post hoc test.

activation and migration[45], whereas in prostate cancer it was found to induce migration of M1-type differentiated macrophages[46,47]. The role of PEDF on the OC immune micro-environment has not previously been investigated.

Several studies have shown that omental macrophages play important roles in the dissemination of ovarian cancer, such as colonization to the omentum and promotion of dissemination potency[26,27]. Ovarian cancer cells are thought to initially colonize the omentum and then disseminate into the peritoneal cavity. Our analysis of the number of surviving OC cells in the peritoneal cavity showed a rapid decrease until day 15, but they increase thereafter. This finding reflects the colonization of tumor cells to the omentum and their subsequent exfoliation from the omentum (Supplementary Fig. 5a). Overexpression of PEDF increased the number of surviving OC cells not only before colonization but also after colonization. Since omentecomy did not significantly

affect the dissemination in PEDF-overexpressed cells (Fig. 3e), the function of PEDF on macrophages is not limited to omental residential macrophages.

IL-10 is an immunosuppressive cytokine produced by M2-like macrophages. We have shown that PEDF increased the IL-10 production in the peritoneal macrophages and intraperitoneal concentration of IL-10. Innate immunity in the retina has been shown to be based on the PEDF-induced production of IL-10 by macrophages[48]. PEDF was also found to increase IL-10 protein and mRNA in macrophages in vitro[49]. We found that the IL-10 level in ascites fluid was significantly higher for patients with serous OC than for those with LMP ovarian tumors, consistent with a previous report of high levels of IL-10 in ascites fluid of OC patients[50]. Our analysis of TCGA data also revealed a worse prognosis for OC patients with higher levels of *IL10* expression (Supplementary Fig. 5b). *IL10* expression in OC tumors was also

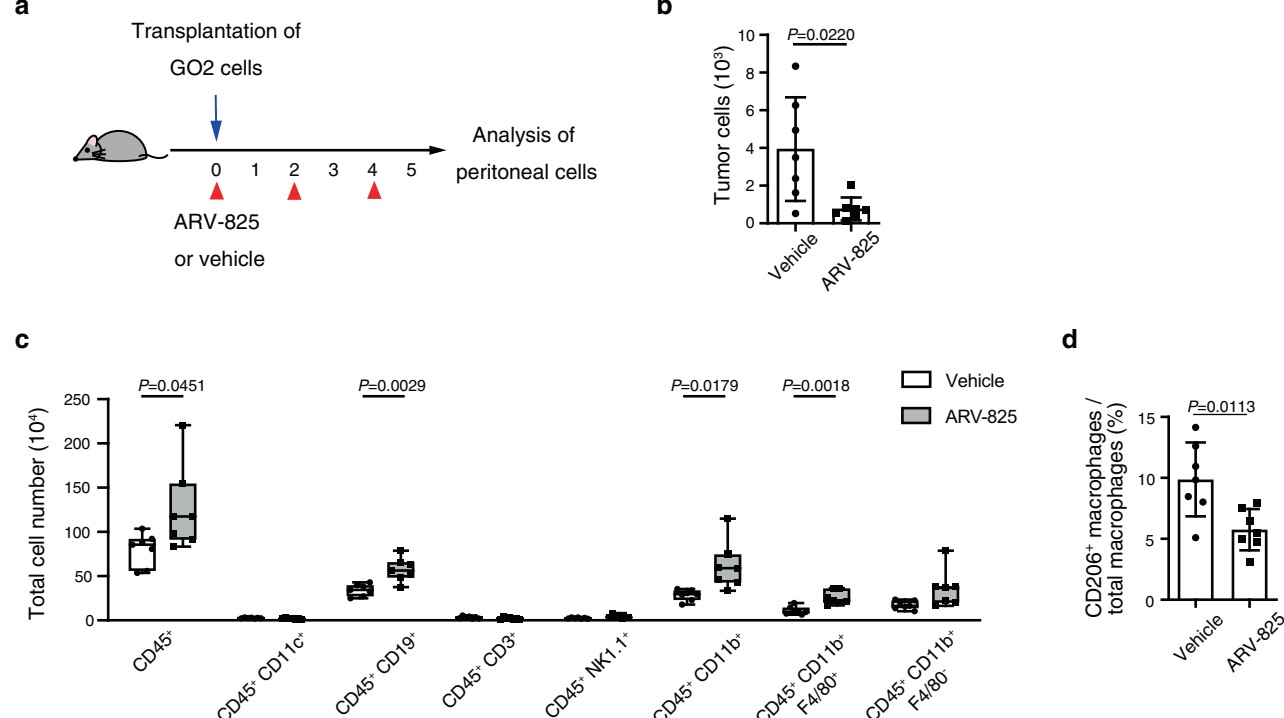

**Fig. 8 Treatment with ARV-825 suppresses OC cell survival and blocks induction of CD206$^+$ macrophages in the peritoneal cavity. a** C57BL/6 J mice were injected i.p. with GO2 cells and treated i.p. with either vehicle or ARV-825 as indicated. **b** Flow cytometric quantitation of GFP$^+$ (tumor) cells in the peritoneal cavity of mice at 5 days after cell transplantation and treatment with vehicle or ARV-825 ($n = 7$ mice each) as in **a**. **c** Flow cytometric analysis of CD45$^+$ cells, CD45$^+$CD11b$^+$ myeloid cells, CD45$^+$CD11b$^+$F4/80$^+$ macrophages, and CD45$^+$CD11b$^+$F4/80$^-$ cells in the peritoneal cavity of mice at 5 days after cell transplantation and treatment with vehicle or ARV-825 ($n = 7$ mice each) as in **a**. **d** Percentage of CD206$^+$ macrophages among total macrophages in the peritoneal cavity of mice at 5 days after cell transplantation and treatment with vehicle or ARV-825 ($n = 7$ mice each) as in **a**. All data are means ± SD and were analyzed by unpaired $t$ test with Welch's correction.

correlated with that of genes associated with the presence of myeloid cells (*CSF1R*, *CD14*, and *CD68*) or macrophage polarization (*MSR1*, *MRC1*, and *CD163*) (Supplementary Fig. 5c). We also detected a strong correlation between expression of the PEDF gene and that of *IL10*, *CD163*, *MSR1*, and *TGFB1* in OC with the use of TCGA data, consistent with our finding that PEDF promotes IL-10 production by immunosuppressive macrophages. The concentrations of PEDF and IL-10 in ascites fluid of OC patients were also correlated. Our data thus have uncovered a role of PEDF in peritoneal dissemination in patients with OC.

An immunosuppressive microenvironment contributes to primary tumor growth[51], invasion[52], metastasis[53], and resistance to chemotherapy[54]. Therefore, the identification of immune-related biomarkers of the tumor microenvironment is vital. We have here shown that the serum and ascites concentration of PEDF were significantly increased in OC patients and were correlated high recurrence rate within 2 years. These findings suggest that the serum level of PEDF may reflect immunosuppressive activity in the peritoneal cavity and can be a useful biomarker to identify patients in need of more intensive treatment.

Finally, we showed that BET inhibitors induced marked downregulation of PEDF expression in a concentration-dependent manner. Treatment with BET inhibitor ARV-825 significantly attenuated tumor cell survival and reduced the number of CD206$^+$ immunosuppressive macrophages in the peritoneal cavity in vivo. ARV-825 was developed recently with the use of PROTAC (proteolysis-targeting chimera) technology and consists of the small-molecule drug OTX015 conjugated with the E3 ubiquitin ligase cereblon[55]. OTX015 has shown high efficacy with tolerable toxicity in phase I clinical trials for patients with solid

tumors or hematologic malignancies[56–58]. Thus, targeting PEDF-facilitated peritoneal dissemination with BET inhibitors is a potential therapeutic strategy for patients with OC.

In summary, we have shown that PEDF increases IL-10–producing immunosuppressive macrophages and thereby plays a key role in the dissemination of OC cells in the peritoneal cavity. PEDF in ascites and serum were elevated in OCs and could be a novel predictor of prognosis for patients with OC. We have also identified BET inhibitors as candidate drugs for the targeting of PEDF expression. Taken together, our data suggest that interference with BET protein-PEDF-IL-10 signaling may provide clinical benefit to patients with OC.

## Methods

**Cell culture**. ID8 cells were obtained from K. F. Roby (University of Kansas Medical Center) and were cultured in DMEM supplemented with 4% FBS as well as insulin (5 μg/ml), transferrin (5 μg/ml), and sodium selenite (5 ng/ml) (ITS; Sigma-Aldrich, St. Louis, MO). SKOV3ip1 cells were obtained from D. Yu (University of Texas, MD Anderson Cancer Center) and were cultured in DMEM-F12 supplemented with 10% FBS. p53-def-MOSE and T-Ag-MOSE cells were obtained from JCRB Cell Bank (Ibaraki, Japan) and were cultured in DMEM supplemented with 10% FBS. RAW 264 cells were obtained from RIKEN Cell Bank (Tsukuba, Japan) and were cultured in MEM supplemented with 0.1 mM NEAA and 10% FBS. All cells were maintained under 5% $CO_2$ at 37°C.

**Viral vectors**. ID8 cells were subjected to retroviral transduction with the GFP expression plasmid pMXs-IRES-GFP (kindly provided by T. Kitamura, The University of Tokyo). Infected cells were sorted with the use of a MoFlo XDP instrument (Beckman Coulter, Miami, FL) to collect GFP$^+$ ID8 (ID8G) cells. Human or mouse PEDF cDNA (Dharmacon, Lafayette, CO) was cloned into the retroviral plasmid pMXs-IRES-puro (kindly provided by T. Kitamura) for retroviral transfer to SKOV3ip1 or ID8G cells, respectively. Plat-E or GP2-293T cells were transiently transfected with the resulting plasmids with the use of the Fugene

HD reagent (Promega, Madison, WI). The empty pMXs-IRES-puro vector was used as a control. Infected cells were subjected to selection in the presence of puromycin (10 μg/ml). We obtained shRNAs that target mouse PEDF mRNA— shPEDF #1 (TRCN0000313899) and shPEDF #2 (TRCN 0000313900)—in the pLKO-puro vector from Sigma-Aldrich. Lenti-X 293 cells (Takara Bio, Tokyo, Japan) were transfected with these constructs with the use of Fugene HD (Promega), and the lentivirus-containing culture supernatants were collected for infection of GO2 cells. GO2 cells stably expressing the shRNAs were selected by culture in the presence of puromycin (10 μg/ml).

**Animal experiments**. Animal care and procedures were performed in accordance with the guidelines of Keio University. Female immunocompetent (C57BL/6 J) and immunodeficient (BALB/c-nu/nu) mice at 6 to 8 weeks of age were obtained from Charles River Japan (Atsugi, Japan). Mice were killed with the use of isoflurane before becoming moribund. All animal experimental protocols were approved by the Keio University Ethics Committee for Animal Experiments.

**Analysis of tumorigenicity**. ID8G, GO2, ID8G-EV, or ID8G-PEDF cells ($5 \times 10^6$) were injected i.p. into 7- to 9-week-old female C57BL/6 J mice, and sham-operated or omentectomized 8-week-old female C57BL/6 J mice, and SKOV3ip1-EV or SKOV3ip1-PEDF cells ($1 \times 10^7$) were injected i.p. into 7- to 9-week-old female BALB/c-nu/nu mice. Mice were killed at the indicated times after cell injection, and the weight of omental tumors, the number or weight of disseminated tumors in the peritoneal cavity, and the volume of ascites fluid were measured.

**Generation of GO2 cells, SKOV3ip1-Om2 cells, and ID8G-as2 cells**. ID8G cells ($5 \times 10^6$) were suspended in 300 μl of PBS and then injected i.p. into 7- to 9-week-old female C57BL/6 J mice. The omentum was isolated between 90 and 120 days after cell injection and was dissociated by incubation for 1 h at 37 °C with collagenase (300 U/ml). Red blood cells in the tissue digest were lysed by exposure to $NH_4Cl$. The single-cell suspension thus obtained was subjected to isolation of $CD45^-GFP^+$ cells with a MoFlo XDP instrument (Beckman Coulter) and cultured for several days. The resulting GO cells were then similarly injected into recipient mice, and the $CD45^-GFP^+$ cells subsequently isolated from the omentum were designated GO2 cells. SKOV3ip1 ($1 \times 10^7$) cells were injected into 6- to 8-week-old female BALB/c-nu/nu mice. Cells isolated from the omental tumors was subjected to isolation of $EPCAM^+$ cells with a MACS instrument (Miltenyi Biotec) and were subsequently recycled with i.p. injection. $EPCAM^+$ cells from the resulting omental tumors were designated SKOV3ip1-Om2 cells. Ascites of mice injected with ID8G cells were collected. After lysis of red blood cells in the ascites, $CD45^-GFP^+$ cells were obtained using MoFlo XDP instrument. The resulting ID8G-as cells were then injected into recipient mice and the cancer cells in the ascites were collected and termed ID8G-as2 cells.

**RNA extraction and RT-qPCR analysis**. Total RNA was extracted from cells with the use of the TRIzol reagent (Invitrogen, Carlsbad, CA) and was subjected to RT with the use of ReverTra Ace qPCR RT Master Mix (Toyobo, Osaka, Japan). The resulting cDNA was subjected to qPCR with TB Green Premix Ex Taq II (Takara Bio). Transcript levels were normalized by the amount of β-actin mRNA. Primer sequences are listed in Supplementary Table 3.

**Immunoblot analysis**. Cells were grown to 80% confluency in six-well plates, after which the growth medium was replaced by OptiMEM (Invitrogen) with or without drugs and the cells were cultured for an additional 24 h. Culture supernatants were mixed with acetone, and the precipitated protein was normalized by cell number and subjected to immunoblot analysis according to standard procedures with primary antibodies to PEDF (Abcam, Cambridge, UK).

**Generation of single-cell clones**. Single ID8G cells were sorted directly with a MoFlo XDP cell sorter (Beckman Coulter) into 96-well plates. After culture for 3–7 days, wells containing a single colony were selected for clone expansion by culture in progressively larger plates.

**Cell proliferation assay**. ID8G-EV or -PEDF cells, GO2 sh#1 or sh#2 cells ($1 \times 10^3$ per well), or SKOV3ip1-EV or -PEDF cells ($1 \times 10^4$ per well) were transferred to 96-well adherent tissue culture plates (Corning, Tewksbury, MA). Cell proliferation was assayed with the use of a CellTiter-Glo Luminescent Cell Viability Assay Kit (Promega) and a multimode plate reader (EnVision; Perkin-Elmer, Boston, MA) after culture for the indicated times with or without drugs.

**Anoikis resistance assay**. ID8G-EV or -PEDF cells, GO2 sh#1 or sh#2 cells ($1 \times 10^4$ per well), or SKOV3ip1-EV or -PEDF cells ($2 \times 10^4$ per well) were transferred to ultralow attachment 96-well plates (Corning) and cultured for the indicated times before assay of cell survival with a CellTiter-Glo Luminescent Cell Viability Assay Kit (Promega) and an EnVision multimode plate reader (Perkin-Elmer).

**Attachment assay**. ID8G-EV or -PEDF cells, GO2 sh#1 or sh#2 cells, or SKOV3ip1-EV or -PEDF cells ($2 \times 10^4$ per well) in 100 μl of serum-free medium were transferred to 96-well flat-bottom plates coated with fibronectin (Corning). The cells were incubated for 90 min at 37 °C and then washed twice with PBS, after which the attached cells were assayed with a CellTiter-Glo Luminescent Cell Viability Assay Kit (Promega).

**Macrophage differentiation assay**. ID8G-EV or ID8G-PEDF cells ($1 \times 10^5$/ml) were transferred to 100 mm culture dishes and cultured for 24 h, after which growth medium was replaced with OptiMEM (Invitrogen) and the cells were cultured for an additional 24 h. The conditioned medium was then collected and centrifuged at $2500 \times g$ to remove cell debris. Bone marrow was extruded from the femur of killed female C57BL/6 J mice at 6–8 weeks of age, and the collected cells were cultured for 7 days in 12-well plates with macrophage complete medium[59]. The resulting BMDMs were then exposed for 48 h to 1 ml of the conditioned medium derived from ID8G-EV or ID8G-PEDF cells before analysis of gene expression. RAW 264 cells ($1 \times 10^5$/ml) were transferred to 12-well plates and cultured for 24 h, then exposed for 24 h to 1 ml of the conditioned medium derived from ID8G-EV or ID8G-PEDF cells before analysis of gene expression.

**Assay of mouse IL-10**. ID8G-EV or -PEDF cells ($5 \times 10^6$) were injected i.p. into C57BL/6 J mice. The mice were killed 5 days after cell injection, and the peritoneal cavity was washed with 3 ml of PBS. The collected peritoneal fluid was centrifuged at $400 \times g$ to remove cells, and the concentration of IL-10 in the supernatant was measured with a High Sensitivity ELISA Kit for IL10 (Cloud-Clone, Houston, TX). The peritoneal cavity of mice injected with ID8G-EV or -PEDF cells ($5 \times 10^6$) was washed with PBS on day 5 to collect free-floating cells. $CD11b^+$ cells were isolated with a use of EasySep Mouse CD11b Positive Selection Kit II and were transferred to 96-well plates ($3 \times 10^6$/ml) and cultured for 24 h. Growth medium was replaced with a new medium and the cells were cultured for an additional 24 h. The conditioned medium was then collected and centrifuged at 2500×g to remove cell debris. The concentration of IL-10 in the supernatant was measured with an ELISA MAX Standard Set Mouse IL-10 (Biolegend).

**FACS and flow cytometry**. For characterization of immune cells in the peritoneal cavity of C57BL/6 J mice at 5 days after i.p. injection of ID8G-EV or -PEDF cells ($5 \times 10^6$), mice were killed and the peritoneal cavity was washed with PBS to collect free-floating cells. The collected cells were analyzed with an Attune Acoustic Focusing Cytometer (Thermo Fisher Scientific, Rockford, IL). For analysis of LPMs, intPMs, and SPMs, we used CytoFLEX Flow Cytometer (Beckman Coulter). The following labeled monoclonal or polyclonal antibodies, obtained from BioLegend (San Diego, CA) unless indicated otherwise, were used for flow cytometry: anti-CD45 (clone 30-F11), anti-CD19 (clone 6D5), anti-NK1.1 (clone PK136), anti-CD11c (clone N418), anti-CD3ε (clone 145-2C11, Thermo Fisher eBioscience), anti-CD11b (clone M1/70), anti-MHC II (clone M5/114.15.2), and anti-F4/80 (clone BM8). Polyclonal antibodies to mouse CD206 (ab64693) were obtained from Abcam and were detected with Alexa Fluor 647–labeled secondary antibodies (BioLegend) at a 1:200 dilution. All primary antibodies for flow cytometry were diluted 1:200. Antibodies to mouse CD16/32 (clone 93, BioLegend) were routinely applied (at a 1:200 dilution) to block Fcγ receptors before staining. FACS was performed with a MoFlo XDP instrument (Beckman Coulter), and sorted cells were frozen in TRIzol (Invitrogen) for subsequent gene expression analysis.

**In vivo treatment of mouse models**. Rat monoclonal antibodies to IL-10R (1B1.3 A) and rat isotype control antibodies (TNP6A7) were obtained from BioXCell (West Lebanon, NH). The antibodies (250 μg) were injected i.v. into mice 2 h before as well as 24 and 72 h after transplantation of ID8G-EV or ID8G-PEDF cells. For treatment with clodronate or control liposomes (Hygieia Bioscience, Minoh, Japan), the liposomes (75 μl) were injected i.v. into mice 1 day before transplantation of ID8G-PEDF cells. For treatment with AS101 (R&D Systems, Minneapolis, MN), the drug (20 μg) or vehicle was administered i.v. into mice 2 h before and daily for 4 days after transplantation of ID8G-EV or ID8G-PEDF cells. For BET inhibitor treatment, ARV-825 (10 mg/kg; ChemieTek, Indianapolis, IN) or vehicle was administered i.p. to mice on the day of as well as 2 and 4 days after transplantation of GO2 cells.

**In vitro treatment with epigenetic drugs**. GO2 ($3 \times 10^4$ per well) or p53-def-MOSE ($5 \times 10^4$ per well) cells were transferred to 12-well, adherent tissue culture plates (Corning) and cultured for 24 h, after which they were exposed to epigenetic drugs and incubated for an additional 24 h. The cells were frozen in TRIzol (Invitrogen) for subsequent gene expression analysis. Anacardic acid, C646, and MB-3 were obtained from Sigma-Aldrich and JQ1, GSK-J4, and JIB-04 from MedChemExpress (Monmouth Junction, NJ).

**Clinical samples for assay of IL-10 and PEDF**. Human part of the study was approved by the Keio University Ethics Committee for Medical Research. Ascites fluid was collected at the beginning of surgery, serum collected before administration of any treatment, and relevant clinical data were obtained from the Hyogo

Cancer Center Biobank. Samples from patients who had received any prior treatment were excluded. Ascites fluid samples were obtained from 28 patients with OC and 3 patients with LMP ovarian tumors. Serum samples were available for 45 patients with OC, 6 patients with benign epithelial ovarian tumors, and 15 patients with epithelial LMP ovarian tumors. The concentrations of IL-10 and PEDF were measured with an IL-10 Human Uncoated ELISA Kit (Thermo Fisher Scientific) and Human SerpinF1/PEDF DuoSet ELISA (R&D Systems), respectively.

**Microarray analysis**. Samples were processed for microarray analysis at the Core Instrumentation Facility of Keio University School of Medicine. Total RNA was extracted from cells with the use of the TRIzol reagent (Invitrogen) and an RNeasy Mini Kit (Qiagen, Hilden, Germany). Cy3-labeled cRNA probes were synthesized from the total RNA and subjected to hybridization with a SurePrint G3 Mouse GE 8 × 60 K Microarray (Agilent technologies, Santa Clara, CA). Raw intensity data for each experiment were analyzed with GeneSpring GX software (Tomy Digital Biology, Tokyo, Japan).

**Analysis of human GEO dataset and TCGA data**. Human RNA-sequencing data (GSE137237) and microarray data (GSE18521) were obtained from NCBI Gene Expression Omnibus. RNA-sequencing data was analyzed by CLC Gx. CPM (counts per million) was obtained by applying TMM Normalization to the CPM values. TCGA data for patients with ovarian serous cystadenocarcinoma were downloaded from GDC data portal. Gene expression data with associated clinical data, including OS or DFS and clinical stage, were assessed. For analysis of SER-PINF1 expression, samples were divided into two cohorts on the basis of the median expression level.

**Statistics and reproducibility**. For in vitro experiments, all results were confirmed with three independent experiments, with the data of one representative experiment being presented. For experiments using BMDMs, results were confirmed with at least two independent experiments, with the data of one representative experiment being presented. Results are expressed as means ± SD unless indicated otherwise. Comparisons between two groups were performed with the two-tailed unpaired Student's $t$ test and Welch's correction or Wilcoxon matched-pairs signed rank test. Comparisons between multiple groups were performed with the Welch's ANOVA followed by Dunnett's post hoc test. For survival analysis, the Wilcoxon test was applied to TCGA data and the log-rank test to mouse data. ROC curve-determined cutoff values of PEDF levels were optimized for diagnostic sensitivity and specificity. Fisher exact test was applied to assess differences between groups. Correlation analysis was performed with Pearson's correlation coefficient. All statistical analyses were performed with GraphPad Prism 9. (GraphPad Software, San Diego, CA). A $P$ value of <0.05 was considered statistically significant.

**Study approval**. All animal experimental protocols were approved by the Keio University Ethics Committee for Animal Experiments. Human samples were provided by the Biobank of Hyogo Cancer Center, and the human part of the study was approved by the Keio University Ethics Committee for Medical Research. Written informed consent for Biobank-dependent research was obtained from all the participants.

**Reporting summary**. Further information on research design is available in the Nature Research Reporting Summary linked to this article.

## Data availability

Microarray data in this study have been deposited into the NCBI Gene Expression Omnibus (GEO) repository under accession number GSE201275. Human RNA-sequencing data and microarray data that support the findings of this study is available at NCBI Gene Expression Omnibus (http://www.ncbi.nlm.nih.gov/geo) under accession numbers GSE137237 and GSE18521, respectively. Data from the TCGA were downloaded from the GDC data portal: https://portal.gdc.cancer.gov/. Data presented in the main figures are available in the Supplementary Data 2. Uncropped blots are shown in Supplementary Fig. 6. All other data supporting the findings of this study are available either within the body of paper, within the Supplementary information, or are available from the corresponding authors upon reasonable request. The plasmids used in this study are available upon request.

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

## Acknowledgements

We thank T. Kitamura for providing the vectors pMXs-IRES-puro and pMXs-IRES-GFP as well as Plat-E cells; K.F. Roby for providing ID8 cells; D. Yu for providing SKOV3ip1 cells; I. Ishimatsu, S. Hayashi, and K. Sonoda for technical assistance; M. Sato and M. Kobori for help with preparation of the manuscript. This work was supported by Japan Society for the Promotion of Science (JSPS) KAKENHI grant JP19K24054 (to S.U.) and by a Grant-in Aid for JSPS Fellows (to SU).

## Author contributions

S.U. designed and performed experiments, collected data, performed statistical analysis, and wrote the manuscript. T.S. designed experiments, discussed data, and edited the manuscript. E.S. assisted with experiments and edited the manuscript. H.S. supervised the project and edited the manuscript.

## Competing interests
The authors declare no competing interests.
