## [Peer Review File · Communications Biology]

Reviewers' comments:

Reviewer #1 (Remarks to the Author):

In the manuscript "Pigment epithelium-derived factor promotes peritoneal dissemination of ovarian cancer through induction of immunosuppressive macrophages" submitted by Sayaka Ueno et al. the authors propose a mechanism where endogenous expressed PEDF promotes increased survival ovarian cancer cells in the peritoneum by inducing IL10 expression in peritoneal macrophages. Overall, the manuscript is well written and provides evidence that PEDF play a role in dissemination of ovarian cancer. However, based on the provided data, this reviewer is not yet fully convinced by the conclusion that PEDF promotes peritoneal dissemination of ovarian cancer through induction of immunosuppressive macrophages in the peritoneum.

Some concerns

1. Recent studies have shown that ovarian cancer, and in particular the ID8 model, initially seeds in the omentum and only subsequently disseminates (after several weeks) the rest of the peritoneum. (eg: Krishnan V, et al. *Commun Biol.* 2020 Sep 22;3(1):524. PMID: 32963283 and Etzerodt A, et al. *J Exp Med.* 2020 Apr 6;217(4):e20191869. doi: 10.1084/jem.20191869. PMID: 31951251). If the purpose of the study is to identify mechanisms for peritoneal dissemination of ovarian cancer, why generate the GO2 cell line from omentum and not from ascites that is known to harbor the disseminating cells?
2. In figure 2, only clone 1 expressing the highest amount of PEDF seems to show overall increased tumor growth and dissemination into peritoneal cavity. What is the expression level of PEDF in this clone compared to the GO2 cell line and is the induced expression physiologically relevant?
3. Most studies from figure 4 and onwards focuses on the initial survival of tumor cells in the peritoneal fluid. Given the important role of the omentum as an essential site for early seeding and establishment of metastatic ovarian cancer, is this survival then relevant for later dissemination of ovarian cancer cells? As increased survival of PEDF expressing cancer cells in peritoneal fluid in an important part of the overall conclusion of this manuscript, the authors should provide data that allows the reader to judge the individual contribution of tumor growth in both peritoneal fluid and omentum to the overall tumor growth. This could for example be done by using omentectomized mice.
4. When analyzing effect of tumor cells on macrophages the authors use flow cytometry of peritoneal lavage and a simple gating strategy identifying macrophages as CD45+, CD11b+ and F4/80+. However, it is well established that the peritoneal macrophage pool consists of two distinct subsets being the large peritoneal macrophages (LMP) and small peritoneal macrophages (SPM). The LPMs are normally identified as F4/80+, MHCII- whereas SPM are identified as F4/80-, MHCII+. Importantly, upon inoculation of ID8 cells i.p. it has previously been shown that LPMs are replaced with a new subset of tumor associated peritoneal macrophages that do not show the same expression in F4/80 as the tissue-resident LPMs (Goossens P, et al. *Membrane Cholesterol Efflux Drives Tumor-Associated Macrophage Reprogramming and Tumor Progression.* *Cell Metab.* 2019 Jun 4;29(6):1376-1389.e4. PMID: 30930171). In order to fully understand the effect of PEDF expressing cancer cells on peritoneal macrophages, the authors should redo their gating strategy to reflect what is already know in the field.
5. The role of macrophages in tumor development is well documented, and depletion of macrophages using clodronate in figure 4 does not provide support for the hypothesis that PEDF promotes peritoneal dissemination of ovarian cancer through macrophages. It only shows that macrophages are important for survival of ovarian cancer cells early after inoculation. The statement in the text should reflect this limitation of the results.
6. Treatment studies in figure 5 + 7 using anti-IL10R, AS101 and ARV-825 have little relevance for proving an effect of PEDF if the ID8-EV cell line is not included as a control. Consequently, it is vital that the authors include this control in the results.
7. From figure 8 the authors conclude from number of cancer cells in peritoneal lavage 8 days after inoculation that inhibition limits OC progression by depleting CD206+ immunosuppressive macrophages. But what about overall tumor growth. Providing data on overall tumor growth at

endpoint would be much more convincing

8. Lastly, this study completely ignores recent articles describing the interplay between macrophages and dissemination of ovarian cancer cells in the peritoneum. The data presented in this paper should as a minimum be discussed in relation to the the articles mentioned above but also to Krishnan V, et al. *Commun Biol.* 2020 Sep 22;3(1):524. PMID: 32963283

Reviewer #2 (Remarks to the Author):

This manuscript reports pigment epithelium-derived factor (PEDF) which was proved to be anti-inflammatory in other tumors as a promoting factor of OC dissemination, which functions through induction of CD206+ IL-10-producing macrophages. PEDF might serve as a prognostic biomarker as high gene expression in tumors is associated with poor prognosis in OC patients. They also identify a BET protein-PEDF-IL-10 axis as a promising therapeutic target for OC. The novel properties of PEDF in OC reported in this manuscript and its driving function in OC dissemination indicate its prognostic and druggable potential.

The article is well-written, well-organized. However, the narrative logic in this manuscript was not very clear and complete, and further data was needed to match the descriptive contents. There are some mistakes and insufficiencies in the manuscript which should be revised.

1. A major concern is about the identification of PEDF in OC. The differential expression of PEDF was analyzed between ID8G cells and GO2 cells to implicate its potential role in dissemination. To make it better generalize to humans, it would be more convincing through consistent analysis between metastatic and non-metastatic human OC samples in TCGA or other databases.
2. The data in Figure 1e could be translated into vivid graphs to present key information more clearly.
3. Since ID8 cells were derived from C57BL/6 mice and lack any of the frequent mutations in HGSC, SKOV3ip1 cell and its corresponding GO2 cells or primary cultured cells from primary tumor and metastasis should be explored as more solid evidence.
4. In addition to fibronectin, the relation between PEDF expression and E-cadherin/Vimentin expression should also be further studied to better understand the "marginal" function of PEDF in OC cells.
5. The authors turn to macrophage after positive findings in Balb/c-nu/nu mice on line 195. The intermediate exploration or explanations from previous studies are needed to drive the line of macrophage rather than other immune cells in the story.
6. Some previous studies reported the role of PEDF in macrophage polarization. What's the performance of the M1 and M2 macrophages between mice injected with ID8G-PEDF cells and ID8G-EV cells?
7. Liposomal clodronate has a selective inhibitory effect on macrophages, but does it have any inhibitory effect on ovarian cancer itself?
8. The p-value of survival analysis in Figure S3a is needed.
9. There are some careless errors such as spelling mistakes on line 264, that the word "cloud" might be "could".

Responses to the Reviewers

Reviewer #1 (Remarks to the Author)

1. Recent studies have shown that ovarian cancer, and in particular the ID8 model, initially seeds in the omentum and only subsequently disseminates (after several weeks) the rest of the peritoneum. (eg: Krishnan V, et al. *Commun Biol.* 2020 Sep 22;3(1):524. PMID: 32963283 and Etzerodt A, et al. *J Exp Med.* 2020 Apr 6;217(4):e20191869. doi: 10.1084/jem.20191869. PMID: 31951251). If the purpose of the study is to identify mechanisms for peritoneal dissemination of ovarian cancer, why generate the GO2 cell line from omentum and not from ascites that is known to harbor the disseminating cells?

We thank the reviewer for this important comment. As the reviewer pointed out, ovarian cancer (OC) cells in the ascites would be the disseminating cells. According to the reviewer's suggestion, we established two cells from the ascites. Cells isolated from ascites of mice injected with ID8G cells by FACS were termed ID8G-as cells and were subsequently recycled with i.p. injection. Cancer cells isolated from the resulting ascites were designated ID8G-as2 cells. We examined the PEDF mRNA expression in parental ID8G cells, ID8G-as cells, ID8G-as2 cells. A stepwise increase in PEDF mRNA levels was observed, consistent with the findings observed in cells derived from the omental tumor (Fig. 1f). We have added these data to the 'Results' section in the revised manuscript (Supplementary Fig.1a, b).

(Fig.1f)

(Supplementary Fig.1a, b)

We have added the following sentences to the revised manuscript:

Page 7, lines 112–115: “Higher abundance of PEDF mRNA and secretion of PEDF protein in GO2 cells compared with parental ID8G cells were confirmed (Fig.1f). Upregulation of PEDF was also observed in cancer cells isolated from the resulting ascites (Supplementary Fig.1a, b).”

2. *In figure 2, only clone 1 expressing the highest amount of PEDF seems to show overall increased tumor growth and dissemination into peritoneal cavity. What is the expression level of PEDF in this clone compared to the GO2 cell line and is the induced expression physiologically relevant?*

We thank the reviewer for this important question. In accordance with the reviewer’s comment, we first compared PEDF expression in clone 1 cells and GO2 cells. PEDF expression in clone 1 cells was approximately a quarter of that in GO2 cells. To clearly show the differences of PEDF expression levels in each clone cells, parental ID8G cells, and highly metastatic GO2 cells, we have changed the supplementary Fig. 2c.

(Supplementary Fig. 2c)

We next examined the relation between PEDF mRNA expression and the survival time using clone 1, 2, 3, 26, and 27 cells, parental ID8G cells, and GO2 cells. PEDF mRNA expression had a strong negative effect on the survival time of mice inoculated with the

corresponding cells ($R=-0.7782$, $P=0.0393$).

We have added these data to the ‘Results’ section in the revised manuscript (Fig. 2e). Accordingly, data of Fig. 2e and 2f in the original manuscript have been moved to the new Fig. 2f and 2g in the revised manuscript.

(Fig. 2e)

We have added the following sentences to the revised manuscript:

Page 8, lines 151–152: “Expression levels of PEDF and survival time of mice injected with corresponding cells showed significantly negative correlation (Fig. 2e).”

3. *Most studies from figure 4 and onwards focuses on the initial survival of tumor cells in the peritoneal fluid. Given the important role of the omentum as an essential site for early seeding and establishment of metastatic ovarian cancer, is this survival then relevant for later dissemination of ovarian cancer cells? As increased survival of PEDF expressing cancer cells in peritoneal fluid in an important part of the overall conclusion of this manuscript, the authors should provide data that allows the reader to judge the individual contribution of tumor growth in both peritoneal fluid and omentum to the overall tumor growth. This could for example be done by using omentectomized mice.*

This is an important point and we really appreciate this constructive suggestion. As the reviewer pointed out, the omentum plays an important role in the dissemination of OC cells. In accordance with the reviewer’s comment, we examined whether omentectomy affects peritoneal dissemination in ID8G-PEDF cells. We found that omentectomy tended

to decrease the potency of peritoneal dissemination in ID8G-PEDF cells, although the difference was not significant. We believe that these data further strengthen our hypothesis that some factor other than the omentum plays an important role in PEDF-dependent peritoneal dissemination.

We have added these data to the ‘Results’ section in the revised manuscript (Fig. 3e, f). Consequently, the data of Fig. 3e–g in the original manuscript have been moved to the new Fig. 3g–i in the revised manuscript.

(Fig. 3e, f)

We have added the following sentences to the revised manuscript:

Page 10, lines 191–202: “Therefore, we hypothesized that PEDF promotes dissemination via some *in vivo* specific mechanism.

Effect of PEDF on OC cells in vivo. The omentum is considered to be an essential site for cancer cell seeding and contributes to peritoneal dissemination in ovarian cancer^{26, 27}. To determine whether PEDF promotes dissemination via the omentum, we assessed the dissemination potency of ID8G-PEDF cells using omentectomized mice (Fig. 3e). Although the number of tumor nodules in the peritoneal cavity and the amount of ascites tended to be lower in omentectomized mice than in sham-operated mice, the differences were not significant (Fig. 3f). This finding suggests that the contribution of not only omentum but also additional factors such as peritoneal fluid to the PEDF-mediated peritoneal dissemination.”

4. When analyzing effect of tumor cells on macrophages the authors use flow cytometry of peritoneal lavage and a simple gating strategy identifying macrophages as CD45+, CD11b+ and F4/80+. However, it is well established that the peritoneal macrophage

pool consists of two distinct subsets being the large peritoneal macrophages (LPM) and small peritoneal macrophages (SPM). The LPMs are normally identified as F4/80⁺, MHCII⁻ whereas SPM are identified as F4/80⁻, MHCII⁺. Importantly, upon inoculation of ID8 cells i.p. it has previously been shown that LPMs are replaced with a new subset of tumor associated peritoneal macrophages that do not show the same expression in F4/80 as the tissue-resident LPMs (Goossens P, et al. *Membrane Cholesterol Efflux Drives Tumor-Associated Macrophage Reprogramming and Tumor Progression. Cell Metab.* 2019 Jun 4;29(6):1376-1389.e4. PMID: 30930171). In order to fully understand the effect of PEDF expressing cancer cells on peritoneal macrophages, the authors should redo their gating strategy to reflect what is already know in the field.

This is a very important point and we really appreciate this constructive suggestion. In accordance with the reviewer’s suggestion, we analyzed peritoneal macrophages using not only F4/80 but also MHC II, to clearly distinguish the subsets in the peritoneal cavity. We found that most CD11b⁺ cells were F4/80⁺ MHC II⁻ LPMs in our experiments. Our experiments were conducted 5 days after inoculation of ID8 cells. The previous study mentioned by the reviewer showed that LPMs were the dominant subset and represented approximately 80% of the macrophages 5 days after inoculation of ID8 cells. Our result is similar to this previous study’s result.

In the revised manuscript, we have added these data to the ‘Results’ section, and changed the Fig. 5a and Supplementary Fig. 4a.

(Fig. 5a)

(Supplementary Fig. 4a)

We have added the following sentences to the revised manuscript:

Page 12, lines 253–263 “Peritoneal macrophages mainly consist of two distinct subsets. One subset is F4/80⁺ MHC II⁻ large peritoneal macrophages (LPMs), which are most abundant, and the other is a minor population of F4/80⁻ MHC II⁺ small peritoneal macrophages (SPMs)³¹. Additionally, a previous study showed that the inoculation of cancer cells induced gradual replacement of LPMs with F4/80^{int} MHC II^{int} macrophages (intPMs)³². Therefore, to clarify which subset is the major macrophage population in our experiments, we assessed the expression of F4/80 and MHC II in peritoneal CD11b⁺ cells. F4/80⁺ MHC II⁻ LPMs were the major population and represented approximately 70% of peritoneal macrophages both in mice inoculated with ID8G-EV cells and ID8G-PEDF cells (Fig. 5a, Supplementary Fig. 4a).”

We next assessed CD206 expression in each subset (F4/80⁺ MHC II⁻ LPMs, F4/80^{int} MHC II^{int} intPMs, F4/80⁻ MHC II⁺ SPMs) and found that the ratio of CD206-expressing cells was increased in F4/80⁺ to int macrophages (LPMs and intPMs) but not in F4/80⁻ MHC II⁺ SPMs.

In the revised manuscript, we have added these data to the ‘Results’ section (Fig. 5b, Supplementary Fig. 4b). Accordingly, data of Fig. 5b-g in the original manuscript have been moved to new Fig. 5c, e-i in the revised manuscript.

(Fig. 5b)

(Supplementary Fig. 4b)

We have added the following sentences to the revised manuscript:

Page 12, lines 266–269 “PEDF expression in the transplanted cells was associated with an increase in the percentage of CD206⁺ cells in F4/80⁺ to int⁺ macrophages (LPMs and intPMs) but not with F4/80⁻ SPMs (Fig. 5b, Supplemental Fig. 4b).”

5. *The role of macrophages in tumor development is well documented, and depletion of macrophages using clodronate in figure 4 does not provide support for the hypothesis that PEDF promotes peritoneal dissemination of ovarian cancer through macrophages. It only shows that macrophages are important for survival of ovarian cancer cells early after inoculation. The statement in the text should reflect this limitation of the results.*

We thank the reviewer for pointing this issue out and the opportunity to correct any misleading parts of the manuscript. The paragraph regarding the experiments of macrophage depletion using clodronate has been modified as follows:

Page 12, lines 232–241: “We showed that PEDF plays a critical role in OC cell survival in the peritoneal cavity at the early stage of dissemination (Fig.3g-i). Next, to clarify the relation between PEDF expression in OC cells and macrophages at the early stage of dissemination, we examined the effect of macrophage depletion by intravenous injection of liposomal clodronate before transplantation of ID8G-PEDF cells (Fig. 4c). Liposomal clodronate itself did not affect cell survival of ID8G-PEDF cells (Supplementary Fig. 3g). The depletion of macrophages resulted in significant attenuation of peritoneal tumor cell survival on day 5, suggesting the supportive role of macrophages on tumor cell survival in the peritoneal cavity at the early stage of dissemination (Fig. 4d).”

6. *Treatment studies in figure 5 + 7 using anti-IL10R, AS101 and ARV-825 have little relevance for proving an effect of PEDF if the ID8-EV cell line is not included as a control. Consequently, it is vital that the authors include this control in the results.*

This is a reasonable comment by the reviewer. In response to the reviewer’s comment, we performed the treatment experiments using anti-IL10R and AS101 in mice injected with ID8G-EV cells. Treatment with anti-IL10R and AS101 showed no significant effects

on ID8G-EV cell survival in the peritoneal cavity. In the revised manuscript, we have added these data to the ‘Results’ section (Fig. 5i).

(Fig. 5i)

The paragraph regarding the experiments of treatment with anti-IL 10R and AS101 has been modified as follows:

Page 14, lines 290–297: “To determine whether the PEDF–IL-10 axis promotes intraperitoneal OC dissemination, we injected ID8G-EV cells or ID8G-PEDF cells into the peritoneal cavity of C57BL/6J mice after the onset of treatment with either antibodies to the IL-10 receptor (IL-10R) or the IL-10 inhibitor AS101 (Fig. 5h). Treatment with either anti–IL-10R or AS101 resulted in a significant reduction in ID8G-PEDF cell survival in the peritoneal cavity, while no significant effect was observed in ID8G-EV cells (Fig. 5i). These findings suggest that induction of immunosuppressive macrophages by PEDF supports OC cell survival in the peritoneal cavity.”

The treatment experiment using ARV-825 was conducted using GO2 cells (Fig. 8). Gene expression profile of GO2 cells, which was established through *in vivo* passage, was different from that of parental ID8G cells (Fig. 1a and 1e). Therefore, we considered that ID8G cells and ID8G-EV cells could not serve as control cells in this experiment.

7. From figure 8 the authors conclude from number of cancer cells in peritoneal lavage 8 days after inoculation that inhibition limits OC progression by depleting CD206+

immunosuppressive macrophages. But what about overall tumor growth. Providing data on overall tumor growth at endpoint would be much more convincing.

This is a very important point and we really appreciate this constructive suggestion by the reviewer. In accordance with the reviewer's suggestion, we examined the overall tumor growth of mice treated with ARV825. Treatment with ARV825 tended to reduce disseminated tumor nodules in the peritoneal cavity, although the difference was not statistically significant ($P=0.18$). We would like to determine an effective route of drug administration in a future study.

8. *Lastly, this study completely ignores recent articles describing the interplay between macrophages and dissemination of ovarian cancer cells in the peritoneum. The data presented in this paper should as a minimum be discussed in relation to the articles mentioned above but also to Krishnan V, et al. Commun Biol. 2020 Sep 22;3(1):524. PMID: 32963283*

We thank the reviewer for pointing out this important issue and providing the opportunity to further discuss the function of macrophages in peritoneal dissemination of OC cells. We examined the surviving cancer cells in the peritoneal cavity at multiple timepoints after injection of ID8G-EV and ID8G-PEDF cells. We found that the number of surviving OC cells in the peritoneal cavity showed a rapid decrease until day 15, but they increase thereafter. We have added these data to the 'Discussion' section (Supplementary Fig. 5a).

We have added the following paragraph to the 'Discussion' section as follows:

Page 20, lines 416–426: “Several studies have shown that omental macrophages play important roles in the dissemination of ovarian cancer, such as colonization to the omentum and promotion of dissemination potency^{26,27}. Ovarian cancer cells are thought to initially colonize to the omentum and then disseminate into the peritoneal cavity. Our analysis of the number of surviving OC cells in the peritoneal cavity showed a rapid

decrease until day 15, but they increase thereafter. This finding reflects the colonization of tumor cells to the omentum and their subsequent exfoliation from the omentum (Supplementary Fig. 5a). Overexpression of PEDF increased the number of surviving OC cells not only before colonization but also after colonization. Since omentectomy did not significantly affect the dissemination in PEDF-overexpressed cells (Fig. 3e), the function of PEDF on macrophages is not limited to omental residential macrophages.”

(Supplementary Fig. 5a)

Reviewer #2 (Remarks to the Author)

1. A major concern is about the identification of PEDF in OC. The differential expression of PEDF was analyzed between ID8G cells and GO2 cells to implicate its potential role in dissemination. To make it better generalize to humans, it would be more convincing through consistent analysis between metastatic and non-metastatic human OC samples in TCGA or other databases.

The reviewer has made an important point and we really appreciate this constructive suggestion. In response to this comment, we analyzed PEDF expression in 11 matched pairs of primary and metastatic human ovarian cancer samples on the basis of an RNA-sequencing dataset (GSE137237). Ten of 11 samples showed higher PEDF expression in metastatic tumors than the primary tumors. ($P=0.0186$, Wilcoxon matched-pairs signed rank test). We have added these data to the ‘Results’ section in the revised manuscript (Fig. 1k).

(Fig. 1k)

We have added the following sentences to the revised manuscript:

Page 7, lines 126–131: “We further assessed whether *SERPINF1* expression was higher in metastatic tumors than in primary tumors using an RNA-sequencing dataset (GSE137237) which are consisted of 11 matched pairs of tumors. We found significantly higher PEDF mRNA expression in metastatic tumors in the peritoneal cavity than in primary tumors (Fig. 1k). Together, these data confirm the dissemination-promoting function of PEDF not only in mice, but also in human ovarian cancer.”

Page 19, lines 401–403: “Furthermore, an analysis of 11 pairs of primary and metastatic tumors in patients with OC showed a significant increase in PEDF expression in metastatic tumors.”

2. *The data in Figure 1e could be translated into vivid graphs to present key information more clearly.*

We appreciate for this great comment. We have changed Fig. 1e to a volcano plot figure (Fig. 1e).

(Fig. 1e)

3. Since ID8 cells were derived from C57BL/6 mice and lack any of the frequent mutations in HGSC, SKOV3ip1 cell and its corresponding GO2 cells or primary cultured cells from primary tumor and metastasis should be explored as more solid evidence.

This is an important point. In accordance with the reviewer's comment, we generated SKOV3ip1-Om2 cells, which corresponded to GO2 cells. The abundance of PEDF mRNA and secretion of PEDF protein were increased in SKOV3ip1-Om2 cells compared with parental SKOV3-ip1 cells, which is similar to the observations between ID8G and GO2 cells. We have added this data to the 'Results' section in the revised manuscript (Fig. 1f, Supplementary Fig. 1c-e). Accordingly, data of Fig. 1f-h and Supplementary Fig. 1a-e in the original manuscript have been moved to new Fig. 1h-j and Supplementary Fig. 2a-e in the revised manuscript, respectively.

(Fig. 1f)

(Supplementary Fig. 1c-e)

We have added the following sentences to the revised manuscript:

Page 7, lines 112–121: “Higher abundance of PEDF mRNA and secretion of PEDF protein in GO2 cells compared with parental ID8G cells were confirmed (Fig.1f). Upregulation of PEDF was also observed in cancer cells isolated from the resulting ascites (Supplementary Fig.1a, b). To rule out the possibility that elevation of PEDF expression in metastatic OC cells are a mouse-specific phenomenon, using SKOV3ip1 human OC cells, we generated SKOV3ip1-Om2 cells, which corresponded to mouse GO2 cells (Supplementary Fig. 1c). The abundance of PEDF mRNA and secretion of PEDF protein were significantly higher in SKOV3ip1-Om2 cells than in parental SKOV3-ip1 cells, which is similar to the observations between ID8G and GO2 cells (Supplementary Fig. 1d, e).”

4. *In addition to fibronectin, the relation between PEDF expression and E-cadherin/Vimentin expression should also be further studied to better understand the “marginal” function of PEDF in OC cells.*

We thank the reviewer for this important comment. In response to this comment, we analyzed E-cadherin and Vimentin protein expression in PEDF-overexpressing ID8G and

SKOV3ip1 cells, and in PEDF-knockdown GO2 cells. PEDF expression levels were not associated with E-cadherin or Vimentin expression levels. We have added these data to the ‘Results’ section (Supplementary Fig. 3f)

(Supplementary Fig. 3f)

We have added the following sentences to the revised manuscript:

Page 10, lines: 186–190 “Association between peritoneal metastasis, survival, and poor chemoresponse and epithelial-mesenchymal transition (EMT) have been reported in OC patients^{24,25}. Therefore, we assessed the effects of PEDF on EMT-related proteins, E-cadherin and vimentin. Expression of PEDF was not significantly associated with E-cadherin or vimentin expression (Supplementary Fig. 3f).”

5. *The authors turn to macrophage after positive findings in Balb/c-nu/nu mice on line 195. The intermediate exploration or explanations from previous studies are needed to drive the line of macrophage rather than other immune cells in the story.*

We thank the reviewer for this suggestion. We have added the explanation from previous studies to the ‘Results’ section

Page 11, lines 214–221: “A previous study using the ID8 model showed no significant difference in the tumor burden between immunocompetent mice and mice lacking mature T and B cells²⁹. In contrast, the depletion of macrophages promoted tumor growth and shortened the survival time independently of T and B cells. Furthermore, macrophages are reported to be essential in spheroid formation of OC cells in the peritoneal cavity and contribute to the dissemination of OC cells in the ID8 model³⁰. These reports indicate the prominent role of macrophages in OC cell dissemination.”

6. Some previous studies reported the role of PEDF in macrophage polarization. What's the performance of the M1 and M2 macrophages between mice injected with ID8G-PEDF cells and ID8G-EV cells?

This is an important point. We consider that we did not show a clear difference in macrophage function between mice injected with ID8G-PEDF cells and those injected with ID8G-EV cells. In response to this comment, we examined cytokines secreted from macrophages and found that peritoneal macrophages isolated from mice injected with ID8G-PEDF cells secreted significantly greater amount of IL-10 compared with those isolated from mice injected with ID8G-EV cells. IL-10 is the most prominent immunosuppressive cytokine secreted by M2 macrophages. Along with our finding of increased IL-10 concentrations in the peritoneal wash fluid shown in Fig. 5d, we believe that the data described above provide a clearer indication of the function of intraperitoneal macrophages. We have added these data to the 'Results' section (Fig. 5c). Accordingly, data of Fig. 5b-g in the original manuscript have been moved to new Fig. 5d-i.

(Fig. 5c)

We have added the following sentences:

Page 13, lines 272–275: “To evaluate the effects of PEDF on IL-10 production in peritoneal macrophages, we collected peritoneal macrophages from mice by MACS and analyzed their IL-10 production. Injection of ID8G-PEDF cells caused significantly higher IL-10 production in peritoneal macrophages than that with ID8G-EV cells (Fig. 5c).”

7. *Liposomal clodronate has a selective inhibitory effect on macrophages, but does it have any inhibitory effect on ovarian cancer itself?*

We thank the reviewer for asking this question and the opportunity to clarify this point. Because we performed the treatment experiments with liposomal clodronate using mice injected with ID8G-PEDF cells (Fig. 4c), we performed additional analyses to determine whether liposomal clodronate affects cell survival of ID8G-PEDF cells. Treatment with liposomal clodronate or control liposome showed no significant effect on cell survival of ID8G-PEDF cells. We have added these data to the ‘Results’ section (Supplementary Fig. 3g).

(Supplementary Fig. 3g)

We have added the following sentences:

Page 12, lines 237–238: “Liposomal clodronate or control liposome themselves did not affect cell survival of ID8G-PEDF cells (Supplementary Fig. 3g).”

8. *The p-value of survival analysis in Figure S3a is needed.*

We than the reviewer for pointing this out. We have added the *P* value to Fig. S3a in the original manuscript (currently new Supplementary Fig. 5b).

9. *There are some careless errors such as spelling mistakes on line 264, that the word “cloud” might be “could”.*

We thank the reviewer for pointing this out. We have carefully checked the manuscript

and corrected the spelling mistakes. We have also added the explanation of original Supplementary Fig 1.d-e in the 'Results' section as follows:

Page 8, lines 158–159: “The abundance of PEDF mRNA and secretion of PEDF protein were increased in SKOV3ip1-PEDF cells compared with SKOV3ip1-EV cells (Supplementary Fig. 1d, e).”

REVIEWERS' COMMENTS:

Reviewer #1 (Remarks to the Author):

The authors have addressed my concerns in the revision with experimental data and clarification. As such, I recommend the acceptance of the manuscript.